# EVA: An Embodied World Model for Future Video Anticipation

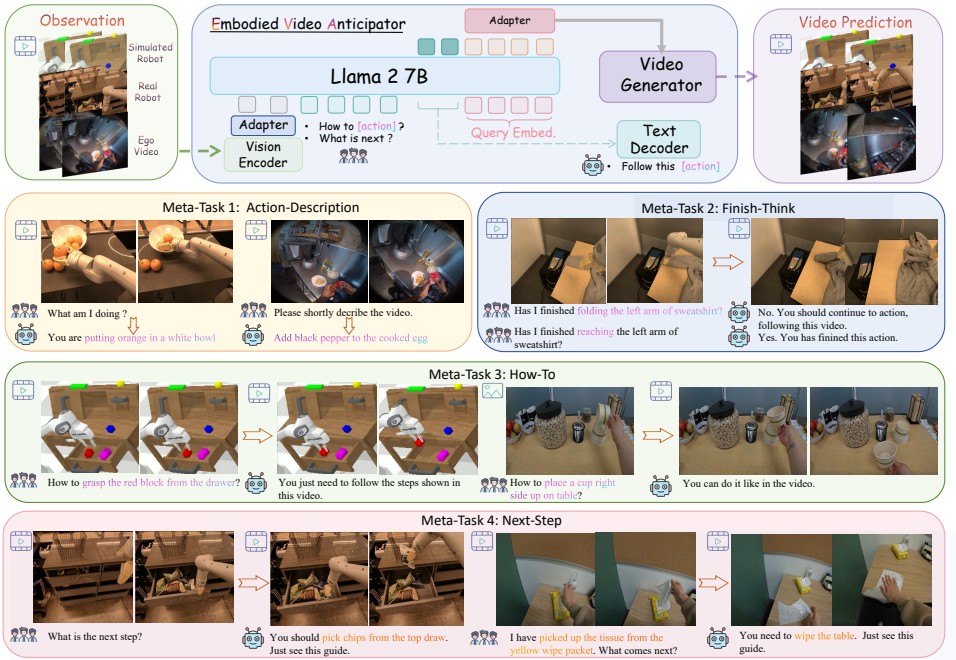

Figure 1: **Meta-tasks of the embodied-video prediction.** We present four meta-tasks, including Action-Description, Finish-Thinking, How-To, and Next-Step, for embodied video anticipation and build the related dataset, benchmark, and model.

## Abstract

World models integrate raw data from various modalities—such as images and language to simulate comprehensive interactions in the world, thereby displaying crucial roles in fields like mixed reality and robotics. Yet, applying the world model for accurate video prediction is quite challenging due to the complex and dynamic intentions of the various scenes in practice. In this paper, inspired by the human rethinking process, we decompose the complex video prediction into four meta-tasks that enable the world model to handle this issue in a more fine-grained manner. Alongside these tasks, we introduce a new benchmark named Embodied Video Anticipation Benchmark (EVA-Bench) to provide a well-rounded evaluation. EVA-Bench focused on evaluating the video prediction ability of human and robot actions, presenting significant challenges for both the language model and the generation model. Targeting embodied video prediction, we propose the Embodied Video Anticipator (EVA), a unified framework aiming at video understanding and generation. EVA integrates a video generation model with a visual language model, effectively combining reasoning capabilities with high-quality generation. Moreover, to enhance the generalization of our framework, we tailor-designed a multi-stage pretraining paradigm that adaptively ensembles LoRA to produce high-fidelity results. Extensive experiments on EVA-Bench highlight the potential of EVA to significantly improve performance in embodied scenes, paving the way for large-scale pre-trained models in real-world prediction tasks. The video demo and benchmark information will be available at https://sites.google.com/view/iclr25-eva.

# 1 INTRODUCTION

A world model integrates raw data from various modalities such as images and language, to imagine how the world evolves as an agent behaves (Ha & Schmidhuber, 2018). It aims to understand the physical world they encounter and generate high-fidelity videos, allowing embodied agents to plan, evaluate, and simulate operations in their own neural space. Such a model shows great potential, particularly in fields like mixed reality, autonomous driving (Gao et al., 2024; Wang et al., 2023b), gaming (Bruce et al., 2024), and robotics (Yang et al., 2023).

Imaging a world model with future prediction ability. It would not only understand environments but also interact by anticipating future actions and generating predictive videos. Such predictive videos could serve as interactive guidelines, akin to a product manual in mixed reality, a driving instructor, or a robot's planning imagination (Du et al., 2023a), greatly enhancing the decision-making process and helping the embodied agents towards Artificial General Intelligence.

This paper aims to enable embodied agents to generate predictive videos using multimodal instructions, allowing them to visualize potential outcomes before taking action. However, existing methods (Zhou et al., 2024) usually focus on conditional simulation and overlook the complexities of multimodal and multi-level time-scale predictions. Moreover, benchmarking such prediction tasks involves several critical steps, data collection, model training, validation, and continuous refinement, which remain largely unexplored. To briefly, there are two main challenges in this task:

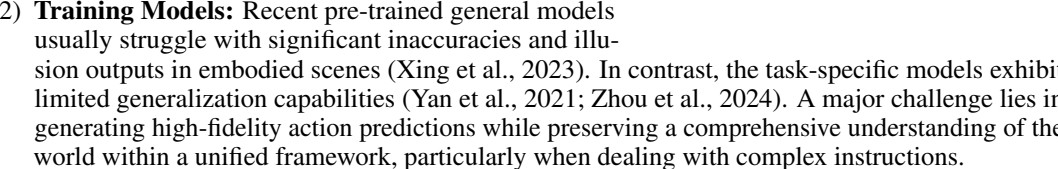

Figure 2: Giving the instruction and observation as input, the world model planning can be decomposed into several meta-questions. With the combination of these meta-questions, **Rethinking**, the world model gives a proper output.

1) **Lack of Benchmark:** Video generation involves producing frame-level actions based on previous motion cues, guided by high-level language instructions (Xing et al., 2024). However, benchmarking these multilevel and multimodal predictions poses challenges due to the complexity of achieving a unified understanding of coarse-grained high-level language instructions.

2) **Training Models:** Recent pre-trained general models usually struggle with significant inaccuracies and illusion outputs in embodied scenes (Xing et al., 2023). In contrast, the task-specific models exhibit limited generalization capabilities (Yan et al., 2021; Zhou et al., 2024). A major challenge lies in generating high-fidelity action predictions while preserving a comprehensive understanding of the world within a unified framework, particularly when dealing with complex instructions.

To benchmark this task, we start by reformulating the problem in a coarse-to-fine manner. Given common sense that frame-level prediction of video generation is a minimum step of prediction, it is not enough for a task-level prediction. As shown in Figure 2, each task-level prediction step requires a comprehensive understanding of the current situation, followed by a next-step prediction of what will happen. Such prediction, in the human mind, could further become pictures or so-called imagination. Then we wonder whether this imagination satisfied this task or not. Such a rethinking process keeps going until this prediction step is completed. Therefore, with the inspiration of recent work (Wei et al., 2022; Liu et al., 2024b; Kawaharazuka et al., 2024), we address these challenges by defining the task within embodied scenes into four meta tasks, *Action-Description, How-To, Finish-Thinking, Next-Step*, as shown in Figure 1. Furthermore, to evaluate such cases, we introduce an embodied video anticipation benchmark (EVA-Bench) to evaluate the video prediction task.

We further propose a novel world model, the Embodied Video Anticipator (EVA), which is capable of generating future prediction videos for embodied scenes, thereby enhancing the interaction experience between humans and machines. To achieve this, we re-collect published datasets from multiple domains to create a diverse dataset and introduce a multi-stage pretraining method. We build an embodied video predictor through a video generation model(VDM) and visual language model(VLM). Additionally, we implement a cross-attention alignment model for end-to-end optimization. We

introduce a few-shot adaptation method with an Ensemble of LoRA Hu et al. (2021) to ensure the model can adapt to specific cases efficiently and incrementally.

In summary, our contributions are as follows:

- **Human-Rethinking-Inspired Task Formulation:** The paper formulates the problem of video prediction by defining it within embodied scenes into four meta tasks: Action-Description, How-To, Finish-Thinking, and Next-Step. This structured approach helps in addressing the complexities of multi-level and multimodal predictions.

- **Embodied Video Prediction Benchmark Creation:** We introduce EVA-Bench, a comprehensive benchmark designed to evaluate the performance of world models in predicting future events within embodied scenes. This benchmark provides a standardized way to measure and compare different models, addressing the challenge of defining and evaluating video anticipation.

- **Embodied Video Anticipator:** We formulate the Embodied Video Anticipator (EVA) framework, which integrates multi-stage pretraining, cross-attention alignment, and few-shot adaptation with an Ensamble-LoRA. Our model can give consistent and long-horizon video predictions.

## 2 RELATED WORK

**Video Generation** With the advent of diffusion-based visual generation models, there has been significant progress in extending the capabilities of video generation. For instance, models like VideoCrafter (Chen et al., 2023; 2024) and VideoPoet (Kondratyuk et al., 2023) have demonstrated impressive abilities in generating high-quality video. Moreover, video generation models with image conditions like Dynamicrafter (Xing et al., 2023), Stable Video Diffusion (Blattmann et al., 2023) and Animatediff (Guo et al., 2023) meet impressive generation quality and have already been used in many areas. Such weakness also happens in some long video generation method (Wang et al., 2023a; Yin et al., 2023). These video-generation models lack reasoning abilities and still struggle with consistency and understanding.

**World Model** World models aim to provide future predictions based on current observations. This concept has been explored in various domains, including Genie (Bruce et al., 2024), which shows interesting ability in gaming simulation, Vista (Gao et al., 2024; Wang et al., 2023b), etc., in autonomous driving. Video prediction is a special world-model-like task, Seer (Gu et al., 2024), AID (Xing et al., 2024) adapting image-to-video generation model to predict the motion of future frames. Additionally, world models such as RoboDreamer (Zhou et al., 2024) and AVDC (Ko et al., 2023) have been utilized as robot simulators. Unisim (Yang et al., 2023), for instance, combining pre-trained web-scale data with embodied videos expands world models' applications. VLP (Du et al., 2023b) integrated language and video generation models for robot planning but stayed at the concept level. A task-level video predictor is still needed.

**Embodied Dataset and Benchmark** On the other hand, such multimodal benchmarks are still missing. Open-X (Padalkar et al., 2023) provides comprehensive robot data, Ego-exo4d (Grauman et al., 2024) provides multi-level language annotation to key steps, and Webvid-10M-Action (Xing et al., 2023) updates the action annotation to the internet video dataset. The MMWorld (He et al., 2024) benchmark includes embodied tasks for visual-question-answer tasks. However, there is still a lack of a video-language-to-video-language dataset that can define and benchmark the prediction ability of the world model.

In summary, world models, whether as simulators or future predictors, require a well-defined task benchmark and model paradigm to advance the field.

## 3 TASK FORMULATION AND BENCHMARK

To benchmark the video prediction task, we first format it. In this section, we will first give a comprehensive task formulation explanation in Section 3.1, and then we will describe how we turn it into four meta-generation problems and how we benchmark them in Section 3.2.

Figure 3: **self-ask (Press et al., 2022) inference pipeline of EVA.** Given the visual observation and human questions as input, EVA would first generate fixed frames of videos and related text answers. Then, the model prompts itself to check the task completion status; if the predicted video is not finished, EVA keeps generating the extended frames until the task completion judgment is true.

## 3.1 TASK FORMULATION

To begin with, we noticed that there is an existing multi-level of language prediction (Cheng et al., 2024); in this paper, we limited the discussion to three levels, frame-level, task-level, high-level, etc. Task-level prediction is direct language instruction, like *"pick up the book"*, and the serial task-level formulate a higher level instruction, e.g., *"Cleaning the table."* Modern video generation models can make frame-level predictions by extending the previous frames.

This paper aims to solve the video prediction problem at the task level. Given common sense that frame-level prediction of video generation is a minimum step of prediction, it is not enough for a task-level prediction. So, the formulation of a task-level prediction and a multi-round of frame-level prediction are required.

The world model function $\mathcal{M}$ takes visual observations $O$ and a question $Q$ as inputs, producing a predicted video $\hat{V}$ and a text response $\hat{A}$ as outputs:

$$(\hat{V}, \hat{A}) = \mathcal{M}(O, Q) \tag{1}$$

To ensure reliable task-level predictions, the model first comprehends the observed video $O$. It then decides whether to provide frame-level or task-level predictions. This decision is transformed into predicted videos $\hat{V}$, which are then iteratively refined until they meet task-level requirements. This process involves four key tasks: Action-Description (demonstrating understanding), How-To (indicating simulation quality), Finish-Thinking (checking task-level generation satisfaction), and Next-Step (showing task-level prediction capability). Only with a solid foundation in these four tasks can the world model make further tasks or higher-level predictions.

## 3.2 EMBODIED VIDEO ANTICIPATOR BENCHMARK

We establish a benchmark for task-level video prediction. Initially, we present a comprehensive design of evaluation metrics that evaluate the video-language to video-language generation task. We've named this the EVA-Score (EVAS). The representation of EVAS is as follows:

$$EVAS \text{normalized} = \sum t \in T \frac{\sum_{i=1}^{n_t} \alpha_{t,i} \psi_{t,i} + \sum_{j=1}^{m_t} \beta_{t,i} \iota_{t,j}}{n_t + m_t} \tag{2}$$

where $T = \{\text{Description, Finish, How-To, Next-Step}\}$ represents the four tasks, $\psi_{t,i}$ is the score of the $i$-th language metric for task $t$, $\iota_{t,j}$ is the score of the $j$-th visual contentment metric for task $t$, and $\alpha$ and $\beta$ denote to the metric specific weight for better normalization. The indices $n_t$ and $m_t$ denote the number of language and visual metrics for each task $t$, respectively.

Therefore, we describe their language and video metrics for each task as follows:

**Action-Description** We expect the text prompt to be short and simple, containing five key elements: subject, verb, object, location, and destination, following the basic English paradigm. We first

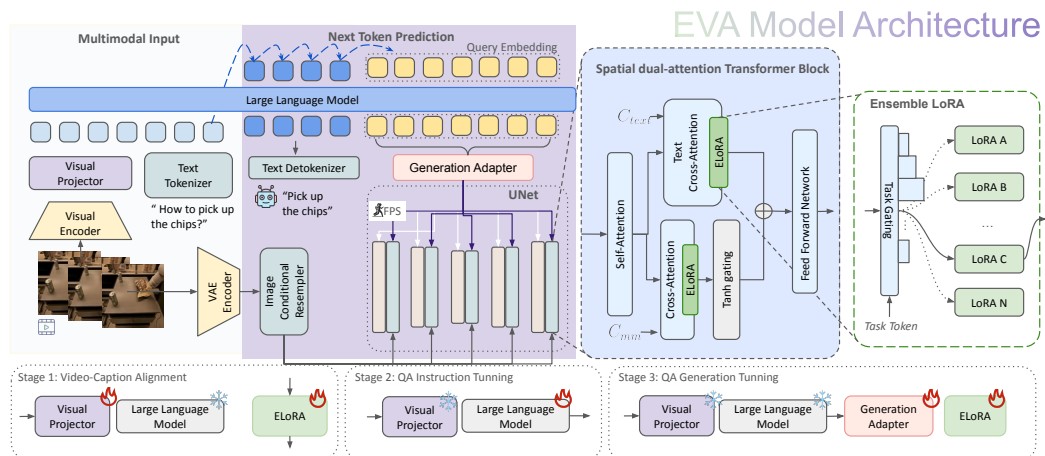

Figure 4: **A unified visual understanding and generation framework of EVA.** The EVA introduces a visual projector in understanding LLM, an image conditional resampler in the generation model, trained a generation adapter as a text condition for denoising UNet, and added an Ensemble LoRA system for domain-specific generation. We train the EVA separately, including three stages of alignment and training.

evaluate this task using a series of text metrics to determine whether the output description closely matches the ground truth. We also added GPT4 (OpenAI, 2024) metric to judge the keywords and CLIP (Radford et al., 2021) score to evaluate the description quality.

**Finsh-Thinking** This task aims to judge whether the frame-level prediction has completed the task. Real-world task generation has dynamic time durations based on different tasks. However, current video generation models can only generate a limited number of frames. In the VQA task, the finish-thinking output "Yes" or "No" could be used to calculate the accuracy. During generation, apart from the existing metric, we put forward a Goal Completion Estimation(GCE) metric to compare the last frame of ground truth and the generation frame.

**How-To** For a "How-To" question, giving the instruction, the world model should be able to turn it into visual output. We introduce several video metrics to formulate the video evaluation metrics, including Motion Smoothness and Multi-stage Consistency inspired by VBench (Huang et al., 2024). These, combined with the aforementioned text metrics, form the correlation metric that represents the quality of "How-To" tasks. The final EVA-Score could benchmark the quality of How-To.

**Next-Step** The last and most important task is to evaluate the task-level prediction of the next step in the video prediction system. We reorganized high-level labeling into task-level Next-Step QA problems to formulate the dataset. While evaluating, we compared the predicted action. We predicted video with the ground truth dataset and obtained the cross-modality correlation among these features, which is also included in our EVA-Score.

These questions help decompose the task into more specific and actionable components, enabling the model to better understand and predict future events across various scenarios. We anticipate that this benchmark and dataset will serve as a foundational setting for evaluating the capabilities of world models. We include the detailed dataset and benchmark information in the Appendix A.5.

## 4 EMBODIED VIDEO ANTICIPATOR

To train the EVA, we designed a multimodal generative large language model, including two main pre-trained models for multimodal prediction. As we described in Section 3, we formulate the problem as a video-language to video-language task, which can be represented as equation 1. EVA uses a multistage training strategy, designed an Ensemble method for domain-specific LoRA Hu et al. (2021) for VDM, interaction tokens to achieve this complex visual prediction task. We describe these key elements in detail in the following section.

### 4.1 VLM BACKBONE

**VLM Backbone** In EVA, we utilize a 7B Visual Language Model (VLM) backbone called ChatU-niVi (Jin et al., 2024). The visual foundation model is CLIP (ViT-L/14) (Radford et al., 2021), while the LLM backbone is the Vicuna-v1.5 (Zheng et al., 2023) model. Moreover, an adaptive parameter-free token clustering algorithm (Jin et al., 2024) is used to reduce the number of video tokens. This backbone setting introduces additional computational overhead and allows training to remain within the 2048-token limit. We first trained the visual project on the Video-Descrtption dataset. We then independently trained the full parameter fine-tuning of LLM on the QA Instruction Tuning dataset, as shown in Figure 4.

**Interaction Token** Inspired by (Peng et al., 2023; Dong et al., 2024), we use special tokens to fit the task better. For image input, the VLM uses an `<image>` token as a placeholder within text tokens. Before being processed by LLM, the `<image>` token is replaced by visual feature tokens, obtained through a visual encoder and visual projector layer. For video inputs, the number of `<image>` tokens used corresponds directly to the number of frames in the video. During generation, we concatenate prefix token `<IMG_P>` with VLM language input as query embeddings to extract the feature. As shown in Figure 4, after obtaining the query embeddings, we use it as a condition for VDM to substitute for text prompt.

### 4.2 VDM BACKBONE

We utilize a 1.5B pre-trained Latent video diffusion model (VDM) called Dynamicrafter (Xing et al., 2023) for the generation model. The VDM conditions on image, fps, and language embedding features to generate a 16-frame video in 2 seconds. It UNet incorporates temporal and spatial transformers, as illustrated in Figure 4, and is pre-trained on a large-scale web video dataset, enabling it to generate dynamic content for open-domain images.

### 4.3 ENSEMBLE OF LORA

We use Low-Rank Adaptation(LoRA) (Hu et al., 2021) to fine-tune the VDM. Let $W$ represent the original weight matrices in the transformer layers. For each domain $d$, we train a low-rank adaptation:

$$W_d = W + \Delta W_d = W + A_d B_d^T \tag{3}$$

where $A_d$ and $B_d$ are low-rank matrices specific to domain $d$. We applied LoRA after each transformer block in video diffusion to quickly adapt the video generation model to different tasks, as shown in Figure 4. Moreover, inspired by the Mixture of Experts (Jacobs et al., 1991), we proposed an Ensamble-LoRA for each domain by a *Task Token* controlled gating system (human-egocentric, real-robot, simulation-robot, etc.):

$$g_d = \text{softmax}(f(\text{Task Token})) \tag{4}$$

where $f$ is a function that maps the task token to gating values, and $g_d$ is the gating value for domain $d$. Therefore, the ensemble output for a given task is computed as:

$$\hat{W} = W + \sum_d g_d \Delta W_d \tag{5}$$

This formulation allows for efficient adaptation across different tasks without discarding previously learned adaptations.

### 4.4 CROSS-ATTENTION GENERATION ADAPTER

Our generation adapter employs a cross-attention module to align the VLM's hidden features with the VDM's text embedding features. Specifically, the adapter first applies a linear transformation to the VLM's output to match the dimensionality of the VDM's feature space. The difficulty loss target trains this adapter to achieve the best generation quality. The detailed module information is introduced in Appendix A.

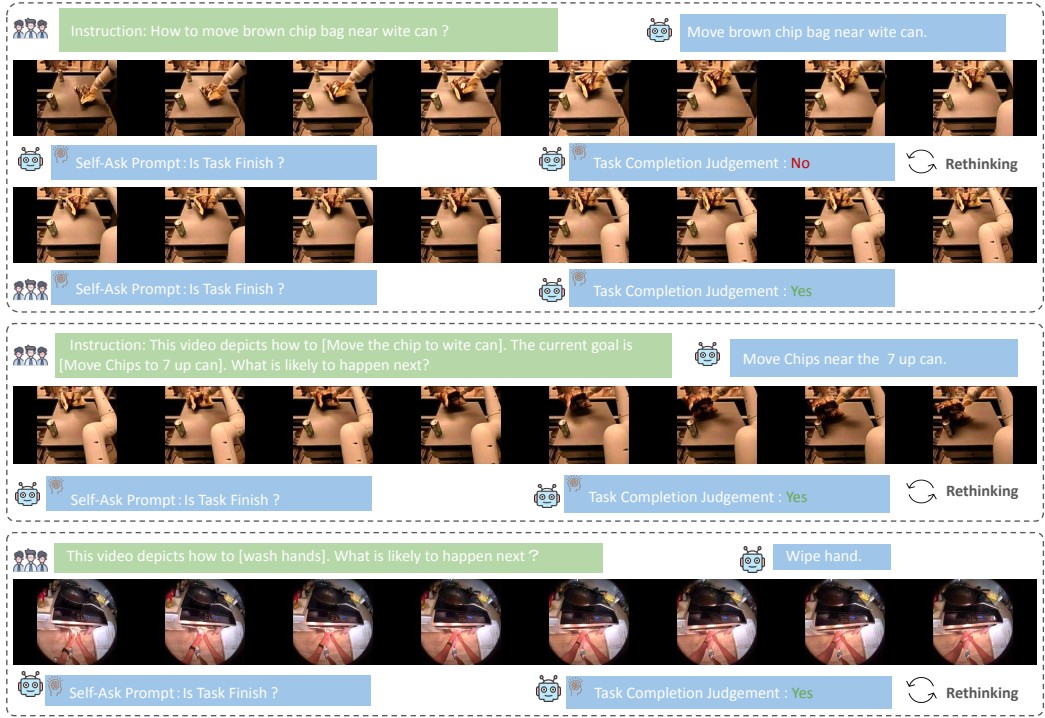

Figure 5: **Visualization results of the How-To, Next-Step, and Finsh-Thinking.** Starting from a random statue, EVA can generate robot motion and human-ego motion according to the instructions. The first two continuous cases show the long-horizon generation ability of EVA; in the last example, EVA can generate video based on its reasoning results. We include more example results on the demo page and in the Appendix B.

## 4.5 MULTI-STAGE TRAINING STRATEGY

To train EVA, we employ a three-stage process. In the first stage, we train the Video-Language Model (VLM) and Video-Domain Model (VDM) separately, and using datasets like COCO (Chen et al., 2015) and a curated subset of CC3M-595K (Sharma et al., 2018) in VLM. The second stage involves aligning the VLM with an instruction tuning dataset, transforming it into an embodied QA bot using multimodal instruction data from sources such as MIMIC-IT (Li et al., 2023), LLaVA (Liu et al., 2023), and VideoChatGPT (Maaz et al., 2024), along with our EVA instruction tuning dataset (introduced in Section 5). In the final stage, we adapt the entire pipeline and train the adapter and VDM Ensemble-LoRA for each task, using the VLM's output hidden embedding as a condition control signal. This stage addresses the diverse visual distribution among EVA-Instruction datasets(EVA-Instruct) by tuning the Ensemble-LoRA and adapter to maximize generation quality. More detailed information on these stages is provided in the appendix and experiment.

## 4.6 SELF-ASK INFERENCE

Last, we propose self-ask inference steps to achieve the embodied prediction task. Given the question and visual observation, we first forward the inference of the VLM and obtain the hidden language state. The hidden state was later forwarded to the generation adapter, which was used as a condition to generate the prediction video. After each round of generation, the model follows the self-ask strategy (Press et al., 2022), which rethinks by asking the model itself, as shown in Figure 3, to check its prediction is a completed video. If the last frame does not satisfy the task completion requirement, the model will autonomously generate the extension of the video based on the previous prediction.

Table 1: **Action-Description results in comparison of VLM.** We compare the open-source VLM models. The QA prompts for each model are included in the appendix. In this table, blue means the best-untrained model.

| Model | BLEU1↑ | BLEU2↑ | METEOR↑ | Rouge-L↑ | CIDEr↑ | Spice↑ | CLIP↓ | GPT-4o↑ |
|---|---|---|---|---|---|---|---|---|
| ChatUniVi-base (Jin et al., 2024) | 0.0969 | 0.0179 | 0.0640 | 0.1497 | 0.0427 | 0.0636 | 27.49 | 9.03 |
| LLAVA-Next-Interleaved (Li et al., 2024b) | 0.0725 | 0.0152 | 0.0741 | 0.1174 | 0.0843 | 0.0982 | 29.63 | 26.94 |
| LLAVA-Next-Video (Zhang et al., 2024) | 0.0717 | 0.0249 | 0.0642 | 0.1062 | 0.1267 | 0.0961 | 30.36 | 25.56 |
| LLAVA-OneVision (Li et al., 2024a) | 0.0874 | 0.0306 | 0.0591 | 0.1118 | 0.2172 | 0.1043 | 27.97 | 22.35 |
| Minicpmv2-6-7b (Hu et al., 2024) | 0.0672 | 0.0164 | 0.0572 | 0.0913 | 0.0404 | 0.0456 | 28.88 | 17.63 |
| qwen2-vl-7b (Yang et al., 2024) | 0.2484 | 0.1643 | 0.1434 | 0.3255 | 0.8914 | 0.2839 | 28.98 | 29.58 |
| GPT-4o (OpenAI, 2024) | 0.2651 | 0.1058 | 0.1671 | 0.2902 | 0.7355 | 0.3015 | **22.96** | 33.19 |
| ChatUniVi-loRA | 0.3007 | 0.1855 | 0.1054 | 0.3268 | 0.8245 | 0.2213 | 24.89 | 31.94 |
| ChatUniVi-Full-Parameter | 0.4105 | 0.1544 | 0.1809 | 0.4416 | 1.9012 | 0.3414 | 25.36 | 38.46 |
| EVA | **0.5735** | **0.5012** | **0.3095** | **0.5873** | **4.0139** | **0.5506** | 24.98 | **62.63** |

## 5 EXPERIMENTS

In this section, we give a comprehensive experiment to evaluate the multimodal understanding and generation ability of EVA on four meta-tasks. First, we assess EVA's VQA reasoning abilities on the Action-Description task in Section 5.1, highlighting the knowledge gaps in existing Vision-Language Models (VLMs) regarding embodied scenes. Next, we provide a detailed comparison of generation performance in Section 5.2, including the video score metrics from EVA-Score on the Finish-Thinking task. In Sections 5.3 and 5.4, we compare EVA against four baselines using EVA-Score to represent the capabilities when facing How-To and Next-Step tasks. Both qualitative and quantitative results are presented throughout, with additional qualitative analyses available in Appendix B. More video demonstrations can be found on the anonymous page at https://sites.google.com/view/iclr25-eva.

**EVA Instruction Tuning Dataset(EVA-Instruct)** It encompasses four tasks featuring videos of humans, real-world robots, and simulated robots. The complete EVA instruction-tuning dataset consists of 500K QA pairs sourced from Open-X-Embodiment (Padalkar et al., 2023), Ego4d (Grauman et al., 2022), Ego-Exo4d (Grauman et al., 2024), and CALVIN (Mees et al., 2022). The EVA-Instruct is presented in a conversational format, paired with single images and videos as visual input. Data sources are summarized and more details are included in Appendix A.5.

**Datasets for EVA-Bench.** EVA-Bench includes a curated collection of 125 high-quality samples from our EVA-Instruct dataset. These samples encompass real-world robots, simulated robots, and egocentric human daily activities. The benchmark is categorized based on meta-tasks. details are included in Appendix D.

**Implementation details.** We set up two kinds of models, EVA-Generator and EVA. EVA-Generator uses Dynamicrafter as the backbone and fully fine-tunes it on the EVA-Instruct. EVA constructs an end-to-end pipeline with fine-tuned ChatUniVi as our VLM backbone and EVA-Generator as our VDM backbone, in the middle, we use a generation adapter to align the feature embedding among two backbones and only train the adapter with the EVA-Instruct.

### 5.1 ACTION-DESCRIPTION VIDEO QUESTION ANSWERING

**Metrics.** First, we present the QA task results from our EVA-Bench. Comparing the BLEU (Papineni et al., 2002), METEOR (Banerjee & Lavie, 2005), ROUGE-L (Lin, 2004), CIDEr (Vedantam et al., 2015), SPICE (Anderson et al., 2016), and CLIP (Radford et al., 2021) scores as multimodal measures. Furthermore, to have a better word analysis, we use GPT-4o (OpenAI, 2024) as an automatic evaluator to obtain a GPT-4o score. The detailed metric description is included in Appendix A.5.

**Main Results.** In Table 1, Qwen2-VL-7B (Yang et al., 2024) and GPT-4o significantly outperform other models in zero-shot inference. However, under GPT-4o's evaluation, while Qwen2-VL-7B remains the best-performing open-source model with a score of 29.58, the gap between the LLaVA-NeXT Liu et al. (2024a) series and Qwen2-VL-7B has noticeably narrowed.

Compared to the zero-shot models, the ChatUniVi (fine-tuned on a 50K subset of our EVA instruction tuning dataset) has better performance across many metrics. Our EVA model remains the top performer, achieving an impressive score of 62.63 under GPT-4o's evaluation. This group of

Table 2: **Finish-Thinking Video Generation Quality Comparison.** Subject Consistency(SC), Background Consistency(BC), Motion Smoothness(MS), dynamic degree(DD), aesthetic quality(AQ), Goal Completion Estimation(GCE), Fréchet Video Distance(FVD)

| Model | Input | SC ↑ | BC ↑ | MS ↑ | DD ↑ | AQ ↑ | GCE ↓ | FVD ↓ |
|---|---|---|---|---|---|---|---|---|
| Dynamicrafter (Xing et al., 2023) | Image+Text | 87.25 | 91.91 | 96.72 | 63.33 | 43.57 | 32.36 | 362.56 |
| Dynamicrafter-Tune | Image+Text | 83.49 | 89.70 | 97.87 | 64.28 | 36.70 | 16.32 | 235.52 |
| EVA-Generator | Image+Text | 95.74 | 95.11 | 99.09 | 50.00 | 41.07 | 14.09 | 177.28 |
| ChatUniVi+Dynamicrafter | Video | 87.10 | 90.82 | 96.97 | 70.00 | **44.38** | 35.78 | 314.11 |
| qwen2-vl-7b+Dynamicrafter | Video | 87.13 | 91.39 | 96.29 | **73.33** | 44.19 | 35.65 | 307.33 |
| ChatUniVi+EVA-Generator | Video | 96.54 | 95.26 | 99.19 | 43.33 | 40.47 | 17.21 | 189.61 |
| qwen2-vl-7b+EVA-Generator | Video | 96.13 | 95.48 | 99.15 | 50.00 | 41.17 | 16.48 | 193,89 |
| LLAVA-OneVision+EVA-Generator | Video | 96.64 | 95.54 | 99.17 | 50.00 | 41.11 | 18.08 | 192.83 |
| EVA-2Stage | Video | 96.68 | 95.82 | 99.17 | 36.66 | 41.30 | **15.58** | 185.89 |
| EVA | Video | **97.11** | **96.01** | **99.31** | 46.67 | 41.72 | 16.04 | **184.81** |

Table 3: **How-To and Next-Step** Task-Level generation evaluation result on the EVA-Bench.

| Task | Model | EVAS-L ↑ | EVAS-V ↑ | EVA-Score ↑ |
|---|---|---|---|---|
| HOW-TO | LLAVA-OneVision+EVA-Generator | 33.81 | 56.39 | 45.10 |
| | qwen2-vl-7b+EVA-Generator | 41.54 | 55.83 | 48.69 |
| | **EVA-2Stage** | 85.51 | 61.57 | 73.54 |
| | **EVA** | 85.51 | 64.21 | **74.86** |
| Next-Step | LLAVA-OneVision+EVA-Generator | 16.75 | 50.73 | 33.74 |
| | qwen2-vl-7b+EVA-Generator | 42.99 | 56.63 | 49.81 |
| | **EVA-2Stage** | 73.02 | 58.85 | 65.94 |
| | **EVA** | 73.02 | 60.68 | **66.85** |

comparisons showcases that the mixture of data we use in EVA outperforms fine-tuning on pre-trained weights, which supports our multi-stage training method.

## 5.2 Finish-Thinking Video Generation

**Metrics** We evaluate the task-condition generation quality of several models. We compare the video generation on Subject Consistency (SC), Background Consistency (BC), Motion Smoothness (MS), Dynamic Degree (DD) (Huang et al., 2024), Fréchet Video Distance (FVD), and especially Goal Completion Estimation(GCE) from task completion evaluation. The quantitative experiments are separated into three groups, as shown in Table 2. On the text domain, Finish-Thinking also includes VQA accuracy comparison on the output of Yes/No, which is not included in this table.

**Main Results** We fine-tune the full UNet of Dynamicrafter on the embodied video dataset named Dynamicrafter-Tune and compare it with the origin Dynamicrafter and EVA-Generator that use Ensamble-LoRA. The EVA-Generator improves significantly in GCE and FVD, with scores of 86.83 and 177.28, respectively. These improvements are due to our effective LoRA design, which successfully adapts the base model to different scenes separately.

Then, we evaluate the video reconstruction ability of the understanding and generation union model. The input video is first converted into a text description by the VLM and then recreated using the first frame and the description. We compare different pairs of VLM+VDM with EVA. Among these frameworks, EVA excels in SC(97.11), BC(96.01), and MS(99.31) and achieves the lowest FVD(184.81) score. This experiment demonstrates the quality of EVA and highlights the efficiency of our multi-stage training and adaption strategy in preserving the generation quality.

## 5.3 How-To Instruction Generation

In Table 3, which demonstrates significant improvements, EVA-2Stage achieved an EVAS-L score of 85.51, an EVAS-V score of 64.21, and an overall EVA-Score of 73.54. Our proposed model, EVA, outperformed all other approaches with an EVAS-V score and an EVA-Score of 74.86. These results highlight that the EVA model, with its multi-stage pretraining and cross-attention alignment, significantly enhances both predictive accuracy and generalization in the How-To task.

Furthermore, by comparing these results, we observed that aligning the output of VLM models to the language instruction format of embodied scenes is essential. Additionally, adopting an end-to-

Figure 6: **Interaction example of "use the kettle."** For each step, input the new observation video, and EVA can generate the instruction video, teaching the user how to use the kettle as an MR handbook.

end approach further enhances the quality of video generation, demonstrating the flexibility and effectiveness of our proposed method. For the qualitative results, we demonstrate how EVA can drive an image with different prompts under real and simulation robots in Appendix B

## 5.4 EMBODIED NEXT-STEP ANTICAPTION

As shown in Table 3, for the quantitative result of the Next-Step task, EVA once again achieved the best performance. As a result, it provides a better text (semantic) condition to guide EVA-Generator for improved generation outcomes. In contrast, LLAVA-OneVision (Li et al., 2024a) and Qwen2-VL-7B performed worse in this task compared to the How-To scenario due to their inability to accurately predict the next-step description. This clearly demonstrates the importance of a VLM that is thoroughly trained in embodied scenes for the Embodied World Model. Such deficiency in EVAS-Language also affects the generation quality, leading to the lower performance of LLAVA-OneVision(50.73) and Qwen2-VL-7B(56.63) on EVAS-Vision. This comparison also shows the overall consistency and quality of EVA-Score.

## 5.5 QUALITATIVE ANALYSIS

In Figure 5, we show the visualization result of our meta tasks and the rethinking. In the first example, given the initial states of the robot and human instruction, we generate the motion "Move brown chip bag near white can." However, such task planning can not be included in 16 frames. With a self-ask rethinking, EVA extends another 16 frames until the chips bag is near the white can. The third demo is an egocentric human QA. Given the previous video and Next-Step question, EVA gives the text response "Wipe hand" together with the extending video.

In Figure 6, we show the potential of using EVA as a handbook in mixture reality. Giving the observation frame and handbook instructions in green color, EVA could keep generating long-horizon videos that guide people on how to use the kettle.

We introduce extensive visualization results, including the action of the following ability in real and simulation robots, more robot Video QA results, and the large dynamic degree of EVA in egocentric human videos in Appendix A.5 and demo page.

## 6 CONCLUSION

In this paper, we introduced the Embodied Video Anticipator (EVA), a unified framework for video understanding and generation in embodied scenarios. By decomposing complex video prediction tasks into four meta-tasks and using the Embodied Video Anticipation Benchmark (EVA-Bench) for evaluation, we demonstrated significant improvements in predicting human and robot actions. EVA's integration of a video generation model with a visual language model, along with a multi-stage pretraining paradigm, enhances its performance in real-world applications. Our extensive experiments highlight EVA's potential to advance large-scale pre-trained models in practical prediction tasks.

**Limitation** In this work, we benchmark the giving embodied video prediction task. However, such benchmarks only consider the interleaved video and language. In embodied scenes, we are expecting to introduce action, sensors, and more modalities into this dataset. Moreover, it will introduce more diverse scenarios, including autonomous driving, human-like robots, etc., in the future.

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

## A   MODEL ARCHITECTURE AND TRAINING

EVA enables the pre-trained diffusion generator and visual language model to provide an autoregressive world prediction model.

### A.1   VISION LANGUAGE MODEL

The Modern Video Language Model (VLM) is based on a Large Language Model (LLM). It leverages the powerful language capabilities of a pre-trained language model to transform the input image or video $I_n$ into latent visual features $\phi$ and then generates a language description output $\tau$. The representation equation of VLM is:

$$\tau = VLM(\phi + \psi) \tag{6}$$

where $\phi = Encoder(I_n)$, and $\psi$ represents the text embedding. Here, $\tau$ is typically in the language embedding sequences, and the visual encoder projects the visual content $I_n$ into the text embedding domain. By this method, VLM trains the visual information in an autoregressive format, similar to another language model.

Given our limited computational resources, it was essential to represent more video frames using fewer video tokens. ChatUniVi's adaptive parameter-free token clustering method significantly reduces the number of video tokens without introducing additional computational overhead, allowing training to remain within the 2048-token limit. We independently trained the VLM on the Embodied-Video-Description dataset, as shown in Fig. 4.

We tested several VLM backbones, including ChatUniVi (Jin et al., 2024), LLaVA-OneVision (Li et al., 2024a), LLaVA-NeXT (Liu et al., 2024a), MiniCPM (Hu et al., 2024), and other mdoels. We found that while existing models were capable of generating detailed descriptions, they struggled with tasks involving prediction and planning, primarily due to the lack of relevant data in their training corpora. Additionally, the simplicity of the text prompts used in the diffusion model's training data necessitated concise responses from the VLM. These responses needed to be composed of short, straightforward sentences that clearly include a subject, verb, object, location, and destination. As a result, we needed to fine-tune the VLMs fully on our dataset.

### A.2   LATENT DIFFUSION MODEL

The Diffusion Model (Ho et al., 2020) is a type of generative model that iteratively refines a noisy input to generate high-quality data samples. It leverages a series of denoising steps to transform an initial noise distribution $\mathbf{z}_0 \sim \mathcal{N}(0, \mathbf{I})$ into a desired data distribution $\mathbf{x}$. The process involves gradually removing noise from the input, guided by a learned model $\epsilon_\theta$ to produce a clear and coherent output. The latent diffusion model(LDM) further uses VAE to scale down the input features and reduce computation costs. Given the initial condition, the representation equation of the LDM (Rombach et al., 2022) is:

$$\mathbf{x}_T = VAE(Unet(\mathbf{z}_0) + \epsilon) \tag{7}$$

where $\mathbf{z}_0$ is the initial noise, $\mathbf{x}_T$ is the final output, *Unet* is the denoising network, and $\epsilon$ represents the text or other control condition embeddings. The Variational Autoencoder (VAE) further decode the latent feature into video.

In our comparisons with Animatediff (Guo et al., 2023), VideoCrafter2 (Chen et al., 2024), and Open-Sora (Zheng et al., 2024), we found that Dynamicrafter (Xing et al., 2023), which employs additional image condition injection methods, excels in retaining low-level features and maintaining high consistency for training-free longer video extensions. The core components of this VDM include a VAE encoder and decoder, an image condition resampler, and a denoising UNet.

## A.3 EVA MODEL ARCHITECTURE

We employ a Vicuna-based VLM and a 3D U-Net architecture VDM to parameterize the EVA model. The model follows the ChatUniVi structure for VLM, and a standard 3D U-Net structure, with a spatial downsampling pass followed by an upsampling pass, utilizing skip connections from the downsampling activations. This process is interleaved with 3D convolution and attention layers. The model and training hyperparameters of EVA are summarized in Table 4 & 5.

Table 4: **Model Architecture for EVA.**

| Name | Type | Parameters |
|---|---|---|
| VDM | UNet | 1.4B |
| VAE Encoder | AutoencoderKL | 83.7M |
| Image adapter | Resampler | 48.8M |
| Text adapter | Resampler | 32.3M |
| VLM | ChatUniVi | 7.0B |
| Query embedding | Linear | 262K |

Table 5: **Hyperparameters for training EVA diffusion model.**

| Hyperparameter | Value |
|---|---|
| Base channels | 320 |
| Optimizer | Adam ($\beta_1 = 0.9, \beta_2 = 0.999$) |
| Channel multipliers | 1, 2, 4, 4 |
| Learning rate | 0.0001 |
| Blocks per resolution | 2 |
| Batch size | 4 |
| Attention resolutions | 4, 2, 1 |
| Num attention heads | 64 |
| Conditioning embedding dimension | 4096 |
| Conditioning embedding MLP layers | 4 |
| Conditioning token length | 64 |
| Dropout | 0.1 |
| Training hardware | 8 Nvidia A800 chips |
| Training steps | 20000 |
| Diffusion noise schedule | cosine |
| Noise schedule log SNR range | [-20, 20] |
| Sampling timesteps | 50 |
| Sampling log-variance interpolation | $\gamma = 0.1$ |
| Weight decay | 0.0 |
| Prediction target | $\epsilon$ |

## A.4 VLM TRAINING DETAILS

During the multimodal pretraining stage, we use image-caption pairs from datasets such as COCO (Chen et al., 2015) and a filtered subset of CC3M (CC3M-595K) (Sharma et al., 2018), curated by LLaVA (Liu et al., 2023). The Visual Language Model (VLM) is pre-trained for one epoch with a batch size of 128, using the AdamW optimizer (Kingma, 2014) and a cosine learning rate schedule. The learning rate is set to 2e-3, with a warmup rate of 0.03.

In the joint instruction tuning stage, we incorporate multimodal instruction data from several sources: (i) the EVA-Instruct dataset, (ii) multimodal in-context datasets like MIMIC-IT (Li et al., 2023), (iii) visual instruction datasets such as LLaVA (Liu et al., 2023), and (iv) video instruction data from VideoChatGPT (Maaz et al., 2024). All input images or frames are resized to $336 \times 336$. Chat-UniVi is trained for two epochs with a batch size of 128 and a learning rate of 2e-5.

**Stage 1: Training VLM and VDM Separately** In the first stage, we train the Video-Language Model (VLM) and Video-Domain Model (VDM) separately to fit them into the embodied prediction

domain. During VLM training, we aim to align the video encoder with a language model using image-caption pairs from various datasets, including COCO (Chen et al., 2015) and a curated subset of CC3M (Sharma et al., 2018) (CC3M-595K) screened by LLaVA (Liu et al., 2023).

**Stage 2: Aligning VLM with Instruction Tuning Dataset** In the second stage, we further align the VLM with the instruction tuning dataset, transforming it into an embodied QA bot. We incorporate open-domain multimodal instruction data from multiple sources: multimodal in-context instruction datasets (MIMIC-IT (Li et al., 2023)), LLaVA visual instruction datasets (Liu et al., 2023), and video instruction data from VideoChatGPT (Maaz et al., 2024). We then add our EVA instruction tuning dataset as described in Section 3. All input images and frames are resized to $336 \times 336$.

**Stage 3: Adapting the Pipeline and Training Ensamble-LoRA** In the final stage, we adapt the entire pipeline and train the adapter and VDM Ensamble-LoRA specifically for each task. Using the output hidden embedding of the VLM as a condition control signal, we employ a Q-former generation adapter to project the feature into VDM's condition input shape. This stage utilizes the EVA instruction tuning dataset. Despite the diverse visual distribution among EVA-Instruction datasets, the language annotations are similar. Under the limited data scale, these domains can negatively affect each other. Therefore, we propose a special tuning method that tunes the Ensamble-LoRA and adapter to maximize generation quality. In this setup, the generation adapter is trained on the whole dataset, while the Ensamble-LoRA is trained on each domain separately. We compare tuning VDM LoRA, CDM full-parameter tuning, and a two-stage inference method in Section 5.

## A.5 TRAINING DATASET

To enhance comprehensive understanding, reasoning, planning and video prediction capabilities in embodied environments, we meticulously curate a comprehensive training dataset including 500K instances, termed EVA-Instruct. This dataset encompasses four tasks, each containing videos of humans, real-world robots, and simulation robots. To enhance the diversity of prompts, we employed ChatGPT-4v (OpenAI, 2024)t to generate question-answer pairs, which were then applied to different tasks. The complete EVA instruction tuning dataset comprises 500K QA pairs collected from Open-X-Embodiment (Padalkar et al., 2023), Ego4d (Grauman et al., 2022), Ego-Exo4d (Grauman et al., 2024), and CALVIN (Mees et al., 2022). The data sources are shown as Table 6. The text in these datasets can be effectively restructured into components such as subject, verb, object, location, and destination, as illustrated in 10, making it highly suitable for our task requirements. The instructions for each meta-task are as follows.

| | Dataset | # Examples | Weight |
|---|---|---|---|
| **Simulation** | CALVIN  (Mees et al., 2022) | 23k | 0.85 |
| **Real Robot** | RoboVQA  (Sermanet et al., 2024) | 800k | 0.1 |
| | RT-1 data  (Brohan et al., 2022) | 70k | 0.5 |
| **Human activities** | Ego4D  (Grauman et al., 2022) | 3.5M | 0.01 |
| | Ego-Exo4D Keystep data  (Grauman et al., 2024) | 21k | 0.9 |

Table 6: **Dataset name, number of training examples, and mixture weights used for EVA-Instruct.**

**Instructions for action description**. The list of instructions used to briefly describe the video content are shown in Table 11. They present the same meaning with natural language variance. Given the complexity of scene understanding in embodied environments, we aim to simplify the problem by selectively incorporating guidelines into the prompt with a probability of 50%. These guidelines are generated using GPT-4V shown in Table 12.

**Instructions for How-to, Finish-Think, and Next-Step**. The list of instructions used to construct the "How-to" format generation is shown in Table13. The instructions for the Finish-Think meta-task are shown in Table 14. Considering dataset differences, we constructed "next-step" prompts using the instructions from Table 15.

**Data Construction for EVA Meta-Tasks**. As shown in Table 6, the data sources for our four tasks are constructed using distinct instructions to create datasets for each meta-task. The datasets

for the How-To and Action-Description tasks are relatively straightforward. Given that the textual annotations in embodied scene datasets generally follow the structure shown in Table 10, we only needed to extend the prompts using GPT-4o and standardize them into the format of subject, verb, object, location, and destination. Due to the ease of constructing the How-To and Action-Description tasks, we built two large datasets: How-To-200K and Action-Description-200K. For the Finish-Think dataset, our statistical analysis indicates that taking the first 25% of a video provides good examples of unfinished tasks. Additionally, some datasets within RoboVQA already contain questions regarding task completion, allowing for direct conversion. Based on this approach, we constructed the Finish-Think-50K dataset. Finally, for the next-step dataset, we utilized the key step annotations from the Ego-Exo4D dataset. This dataset marks key steps for each segment of a complete video, making it easier to convert into a next-step prediction task. In the Open-X-Embodiment's real robot datasets, such as RoboVQA, which contain long-horizon task annotations involving a sequence of multiple steps, we converted these sequential steps into next-step tasks by focusing on the ordered steps provided. Using this approach, we constructed the Next-Step-50K dataset.

## B EXTENSIVE EXPERIMENTS

In this section, we provided the results of the extent visualization experiment. Figure 7 provides visualization results on simulation robot data, demonstrating that EVA could drive the robot by text instruction. The result shows that with proper action, the following is quite accurate.

Figure 8 and Figure 9 also show the action following generation ability of EVA in real robots. Figure 8 shows EVA could answer the question by generation the video.

Figure 10 is an ego-centric generation example. With proper training, the model has some planning abilities, generating a continuous motion sequence that first turns right and then gets the sugar.

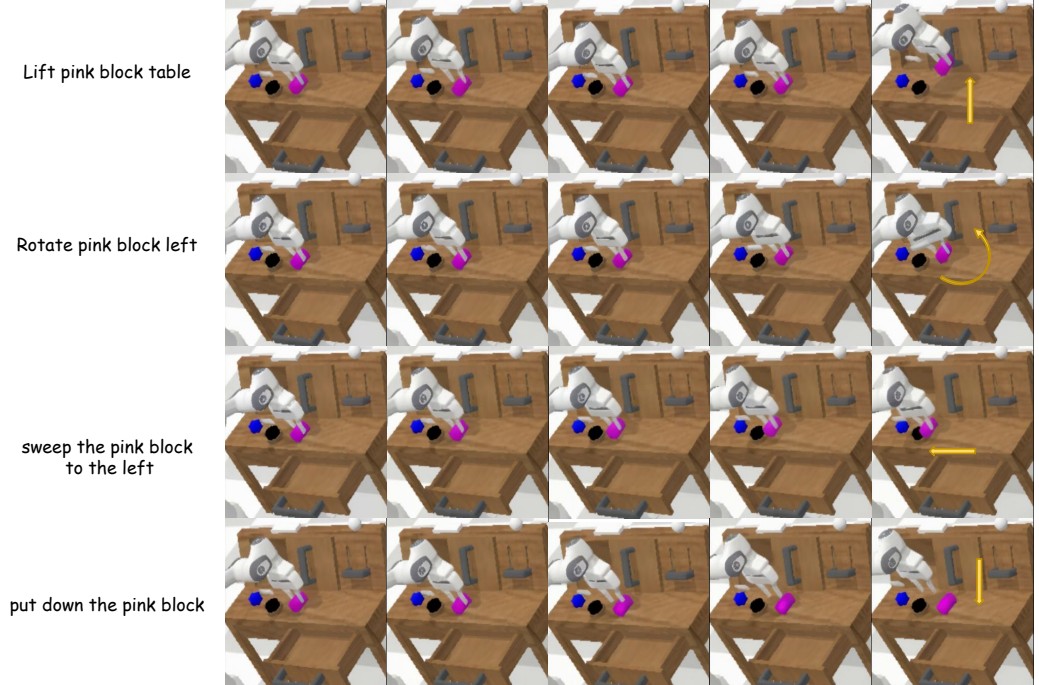

Figure 7: We show the prompt following the ability of the EVA on the simulation robot. Given the same input video, the model can generate different actions according to different instructions.

How would you do [pick green can from middle shelf of fridge]?

Could you explain how to [pick green can]?

How to [pick green rice chip bag]?

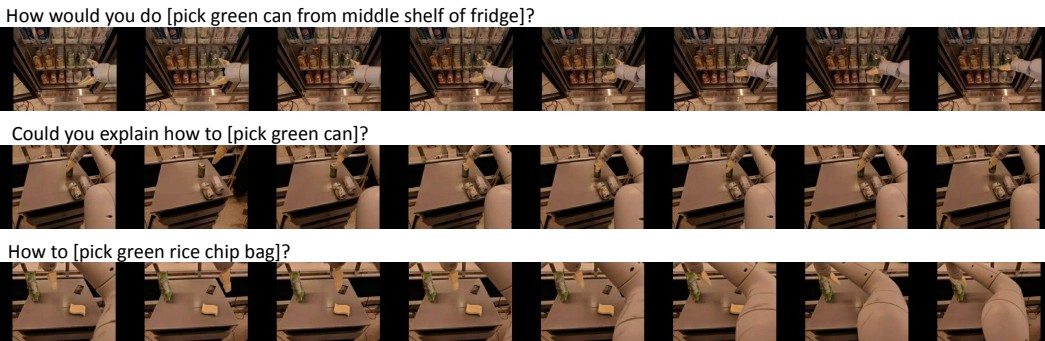

Figure 8: **EVA's action control abilities on the real robot videos.**

knock orange can over

knock redbull can over

move brown chip bag near blue chip bag

move green jalapeno chip bag near apple

move sponge near rxbar blueberry

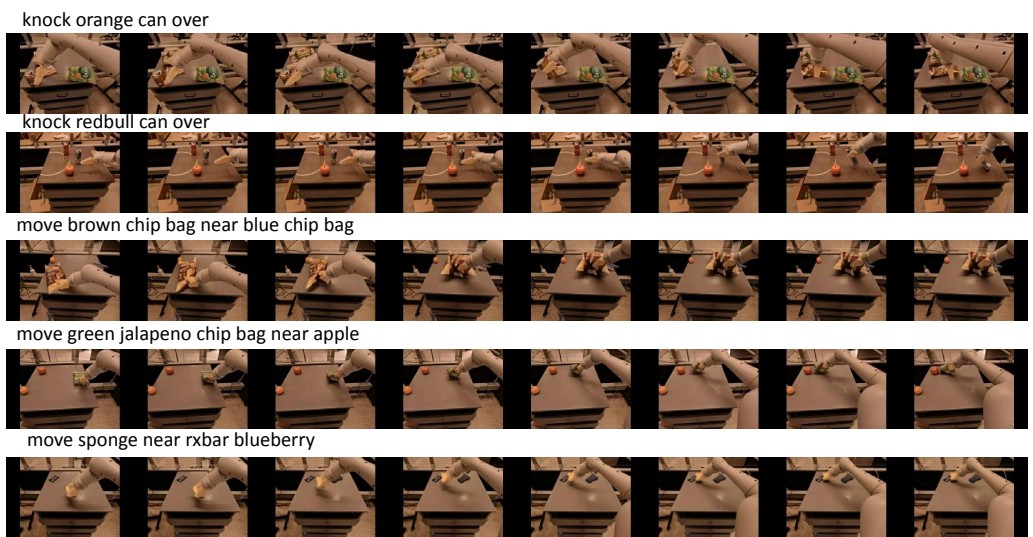

Figure 9: **EVA's action control abilities on the real robot videos.**

## C    COMPARISON IN RT1

We compare the success rates of human evaluation tasks in RT1, following the framework of RoboDreamer[6]. Evaluation spans two groups—seen prompts and unseen prompts—and includes a comparison of AVDC, EVA without finish-thinking, and EVA.

| Model | Pick Object | Move Object Near Object | Place Object Upright | Knock Object Over | Open/close | Place Object into Receptacle | Summary |
|---|---|---|---|---|---|---|---|
| AVDC | 11/16 | 9/12 | 1/2 | 4/4 | 2/8 | 2/8 | 29/50 (58%) |
| EVA (w/o finish thinking) | 13/16 | 12/12 | 2/2 | 4/4 | 4/8 | 6/8 | 41/50 (82%) |
| EVA | 13/16 | 12/12 | 2/2 | 4/4 | 4/8 | 7/8 | 42/50 (84%) |

Table 7: The human evaluation results of RT-1 Brohan et al. (2022). We perform human evaluation results to judge the task completion rate on seen prompts and seen cases of EVA and AVDC.

In seen tasks, we randomly selected 50 tasks from the validation set of RT1, including 6 tasks, across multiple scenes(Pick Object, Move Object Near Object, Place Object Upright, Knock Object Over, Open/close, Place Object into Receptacle). For the seen tasks in Tab. 7, AVDC has a 58% success rate, while EVA is 28% higher in total success rate. Moreover, EVA performed better in the Move Object task with a 100% success rate, showing good prompt-following ability. The Open/close task is especially hard since a few cases like "open right fridge door" include the transparent glass door.

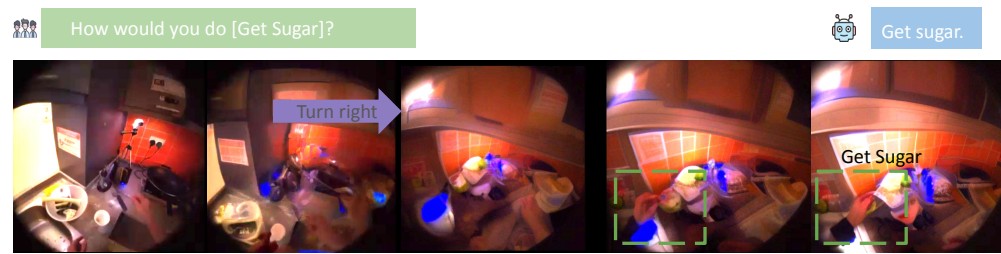

Figure 10: **EVA can do large Motion.** In this example, the generation video first turns left and then gets the sugar with a different hand.

| Model/Tasks | Pick Object | Move Object Near Object | Place Object Upright | Knock Object Over | Open/close | Place Object into Receptacle | Summary |
|---|---|---|---|---|---|---|---|
| AVDC | 2/4 | 1/4 | 1/1 | 1/2 | 1/1 | 1/3 | 7/15 |
| EVA (w/o finish thinking) | 2/4 | 1/4 | 0/1 | 2/2 | 0/1 | 0/3 | 5/15 |
| EVA | 3/4 | 4/4 | 1/1 | 2/2 | 1/1 | 1/3 | 12/15 |

Table 8: The human evaluation results of RT-1 Brohan et al. (2022). We perform human evaluation results to judge the task completion rate on EVA and AVDC unseen prompts.

For unseen tasks, we start from the existing cases and manually change the subject or object of the prompt. For example, "Place coke can into the bottom drawer" to "Please close bottom drawer". AVDC performance is better than EVA (w/o finish thinking) in Place, knock, and Open/Close tasks, since the trajectory is longer than EVA (w/o finish thinking) can generate simultaneously. However, EVA could fix this issue and significantly improve the success rate by extending the video.

# D EVA-BENCHMARK

To facilitate evaluation, the proposed EVA-Bench curated a collection of 125 high-quality samples from our EVA-Instrut, covering real-world robots, simulated robots, and egocentric human daily activities. These samples encompass diverse scenarios such as pick-and-place tasks, cooking, bike repair, COVID testing, and indoor organization. Drawing from existing datasets, we categorize them into three groups: egocentric human videos, real-world robots, and simulations. The statistical distribution of different scenes in our EVA-Bench is shown in Fig. 12.

## D.1 BENCHMARK EXAMPLES

We selected frames from the benchmark in three areas: egocentric human videos, real-world robots, and simulated robots. These frames showcase the richness and diversity of our embodied scenes, as shown in Fig. 11. The first three rows in the figure represent scenes from cooking, COVID testing, and bike repair. The fourth and fifth rows display indoor robotic arms manipulating various objects, while the sixth row shows scenes involving simulated robots.

## D.2 EVA SCORE LANGUAGE METRICS

**BLEU (Bilingual Evaluation Understudy Score):** BLEU1 and BLEU2 represent the BLEU scores using 1-gram and 2-gram matches, respectively. BLEU measures the overlap of n-grams between the generated text and reference text. Higher scores indicate greater similarity to the reference.

**METEOR (Metric for Evaluation of Translation with Explicit ORdering):** METEOR considers factors like stemming, synonyms, and word order, making it more flexible than BLEU. It evaluates translation quality based on precision, recall, and a penalty for longer sentences. Higher scores indicate better translation quality.

**ROUGE-L (Recall-Oriented Understudy for Gisting Evaluation - Longest Common Subsequence):** ROUGE-L measures the quality of text summaries and translations by calculating the longest common subsequence (LCS) between the generated and reference texts. It focuses on recall, with higher scores indicating better coverage of the reference content.

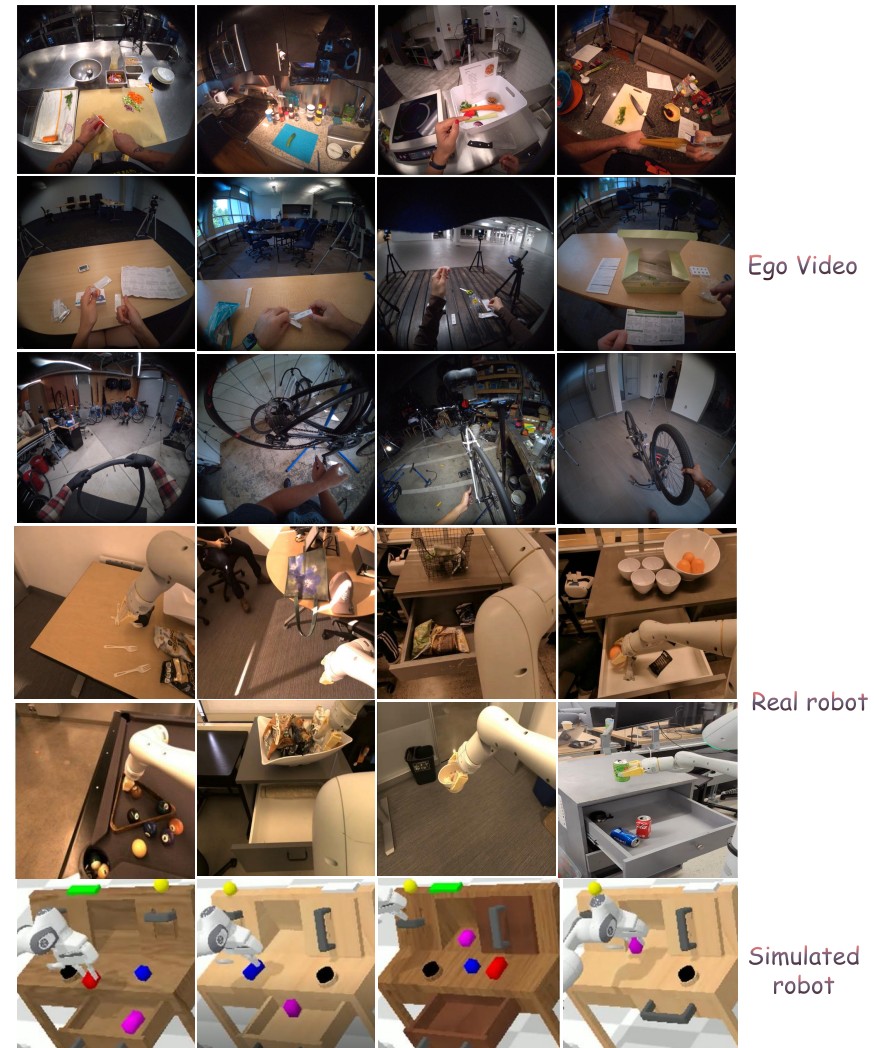

Ego Video

Real robot

Simulated robot

Figure 11: **Random frames from EVA-Bench.**

**CIDEr (Consensus-based Image Description Evaluation):** CIDEr is used primarily for image description tasks. It evaluates the quality of descriptions by calculating the TF-IDF weighted n-gram similarity between the generated and reference descriptions. Higher scores indicate greater consistency with the reference descriptions. While calculating EVAS-Language points, we normalized the CIDEr by:

$$X_{\text{norm}} = X/10$$

where X is the CIDEr score.

**SPICE (Semantic Propositional Image Caption Evaluation):** SPICE evaluates image descriptions by parsing the generated and reference descriptions into semantic graphs. It focuses on the semantic content and relationships within the descriptions. Higher scores indicate better semantic alignment with the reference descriptions.

**CLIP(Contrastive Language-Image Pre-training) score**: CLIPScore is a reference-free evaluation metric for image captioning. Unlike traditional metrics that compare generated captions to reference captions, CLIPScore uses a pre-trained CLIP model to directly measure the similarity between the generated caption and the image itself. This approach leverages the model's ability to understand both images and text, providing a robust evaluation of how well the caption describes the image. Higher CLIPScores indicate better alignment between the image and the generated caption. In EVA

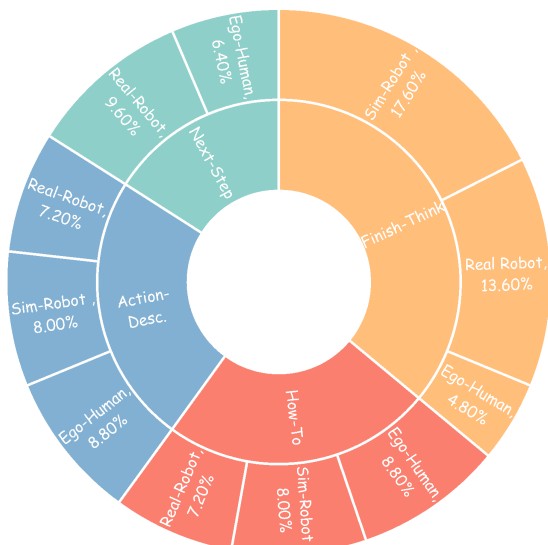

Figure 12: This chart illustrates the distribution of various embodied scenes categories within the EVA-Bench.

Bench, we normalize the CLIP score between 0 1 by:

$$X_{\text{norm}} = \frac{X' - \min(X')}{\max(X') - \min(X')}$$

Where $x'$ is the reciprocal of the $X$, we fix the upper and lower bound of the clip score in 10 50. **CIDEr**, its upper bound is 10.

**GPT-4o as a Judge**: Unlike traditional similarity-based methods, GPT-4o emphasizes semantic understanding. In our implementation, we format the question, model output, and reference into a prompt, as outlined in Appendix D.4, and input it into the GPT-4o evaluator. The comparison between the generated and reference answers is based on four key criteria: object, action type, location, and attribute. During the model evaluation, we observed that some generated responses, despite being semantically close to the ground truth, received low scores. Conversely, responses omitting key information occasionally received high scores. For example, the ground truth might state, "Cut out the tomato stem with a knife on the cutting board" while Qwen2-VL-7B (Wang et al., 2024) predicts, "Chop the tomato with a knife on the cutting board". Despite high BLEU scores, the key difference between removing the stem and chopping the tomato remains significant. Therefore, using GPT-4o as a judge to score QA text pairs and model-generated responses is essential. By leveraging GPT-4o's advanced analytical and reasoning capabilities, we can more accurately evaluate the similarity between generated and reference texts. The specific prompt is detailed in Appendix D.4.

### D.3 EVA Score Video Metrics

The metrics of Overall Consistency, Motion Smoothness, Background Consistency, and Subject Consistency are inspired by the contributions from the open-source project VBench (Huang et al., 2024).

**Overall Consistency:** We also evaluate the overall consistency between video and text using ViCLIP (Wang et al., 2022), which measures how well the generated video aligns with general text prompts in terms of both semantics and style.

**Motion Smoothness:** While Subject Consistency and Background Consistency focus on the temporal consistency of the appearance, Motion Smoothness evaluates whether the motion in the generated video is smooth and adheres to real-world physical laws. This is assessed using motion priors from a video frame interpolation model.

**Subject Consistency:** This metric assesses the alignment between the subject in the input image and the subject in the generated video, also using DINO (Caron et al., 2021) features and order-statistics schemes.

**Background Consistency:** This metric evaluates the coherence between the background scene in the input image and the generated video. It utilizes DINO features and carefully designed order-statistics schemes.

**Background Consistency:** This metric evaluates the coherence between the background scene in the input image and the generated video. It utilizes DINO (Caron et al., 2021) features and carefully designed order-statistics schemes.

**Aesthetic Quality:** This metric refers to the perception of beauty or artistic value in a work, whether it's traditional art, design, or even generative models. It encompasses elements like composition, color, texture, and form and how these elements evoke emotional responses from the viewer (Schuhmann et al., 2022).

**Goal Completion Estimation:** This metric estimates the final frame from the generative model compared to the ground truth. We evaluated the video by DreamSim (Fu et al., 2023) feature. Moreover, we also performed a comparison of DINOV2, CLIP, and DreamSim, as shown in Tab. 9 and used DreamSim last. The greater the GCE, the generated image is to the target. However, evaluating the robot manipulation task and human motion from pixels still remains a challenge, which also left room for future research.

| Model | Input | Dreamsim↓ | DINO↑ | CLIP-I↑ | FVD↓ |
|---|---|---|---|---|---|
| Dynamicrafter (Xing et al., 2023) | Image+Text | 32.36 | 84.67 | 80.96 | 362.56 |
| Dynamicrafter-Tune | Image+Text | 16.32 | 92.35 | 80.72 | 235.52 |
| EVA-Generator | Image+Text | 14.09 | 93.49 | 86.83 | 177.28 |
| ChatUniVi+Dynamicrafter | Video | 35.78 | 82.84 | 80.80 | 314.11 |
| qwen2-vl-7b+Dynamicrafter | Video | 35.65 | 83.18 | 81.96 | 307.33 |
| ChatUniVi+EVA-Generator | Video | 17.21 | 90.95 | 88.48 | 189.61 |
| qwen2-vl-7b+EVA-Generator | Video | 16.48 | 91.58 | 85.87 | 193.89 |
| LLAVA-OneVision+EVA-Generator | Video | 18.08 | 91.29 | 88.74 | 192.83 |
| EVA-2Stage | Video | **15.58** | 91.93 | **90.19** | 185.89 |
| EVA | Video | 16.04 | **92.20** | 89.09 | **184.81** |

Table 9: Comparison of DINO, CLIP-I, DreamSim, and FVD. We compare the last frame of the generated video with the last frame of the ground truth.

### D.4    MODEL INFERENCE PROMPTS

Since the annotated answers in our dataset typically focus on key actions, composed of elements such as subject, verb, object, location, and destination, existing general VLMs struggle to generate responses in the same style. They often produce redundant or irrelevant scene descriptions. To address this, we designed specific prompts to guide the visual language models (VLMs) in generating concise, non-redundant answers. The prompt designs for various VLMs are listed in Table 16. For each meta-task in our EVA-Bench, all models, except for our EVA model, use the same prompts listed in Table 16.

### D.5    EVALUATION PROMPTS

To evaluate general Vision-Language Models (VLMs) in a zero-shot setting, results using standard text evaluation metrics are often poor. Through experimental analysis, we found that traditional metrics like BLEU, METEOR, and ROUGE-L do not effectively capture similarities in actions, objects, locations, and other essential factors. Therefore, we follow the approach proposed by EgoThink (Cheng et al., 2024) and employ GPT-4O to assess the predictions of these VLMs.

However, directly using EgoThink's prompts often leads to extreme scores, with many reasonable predictions being rated as 0. We believe this discrepancy stems from the increased complexity of tasks in embodied scenes, which demands a more nuanced and sophisticated evaluation framework.

For the step description generation task, we guide GPT-4o to assess predictions based on four key criteria: **object**, **action type**, **location**, and **attribute**. Each criterion is scored as follows: 1 if fully

correct, 0.5 if partially correct or somewhat aligned, and 0 if incorrect. The final score is the average of these four criteria.

**Object** evaluates whether the objects involved in the action are correctly identified. **Action type** leverages GPT-4O's reasoning ability to assess whether the predicted action aligns with the ground truth. For instance, comparing "get the fork from the table" and "pick up the fork from the table," the use of "get" and "pick up" would result in a score of 0.5. **Location** assesses whether the location where the action is performed is accurately described, along with the broader context. If the action involves movement (e.g., moving an object from one place to another), the evaluation considers whether the starting point, destination, or path is correctly specified. **Attribute** examines whether the attributes of the objects involved (e.g., size, color, state, condition) are described accurately.

Our designed prompts are presented in Table 17.

> - "C adds the shredded ginger into the small stainless cowl with his right hand."
> - "C steps to the right while holding her head and swaying her hips."
> - "C places his right hand around the waist of woman X."
> - "Tighten the left axle nut with your left hand."
> - "Place the black chip bag in the tray."
> - "Grasp the red block from the drawer."
> - "Keep the brown potted plants together."

Table 10: **The example answers of EVA-Instruct-Answer.** Example answers from EVA-Instruct-Answer. In this context, C refers to the wearer of the egocentric camera, while x represents the other person involved in the interaction.

- "What is happening in this egocentric video?"
- "Can you describe the interaction in this video?"
- "What actions are being performed in this video?"
- "Please provide a description of the activity in this video."
- "What is the person/robot doing in this video?"
- "What object is being interacted with in this video?"
- "Can you summarize the actions in this video?"
- "What task is being carried out in this video?"
- "What is the main activity shown in this video?"
- "Can you provide a brief description of the video content?"
- "What interaction is taking place in this video?"
- "What is the main focus of this video?"
- "What is the subject doing in the video?"
- "Can you explain the actions seen in this video?"
- "What specific steps are being taken in this video?"
- "What is the sequence of actions in this video?"
- "What key activities are being shown in this video?"
- "What is the primary task in this video?"
- "What is the individual engaged in within this video?"
- "What detailed actions are depicted in this video?"
- "Can you outline the main steps in this video?"
- "What is the purpose of the actions in this video?"
- "What are the key interactions in this video?"
- "What process is demonstrated in this video?"
- "What is the sequence of events in this video?"
- "What detailed activities are performed in this video?"
- "What main actions can be observed in this video?"
- "What specific task is being executed in this video?"
- "What are the primary actions taking place in this video?"
- "Can you detail the key steps shown in this video?"

Table 11: **The list of instructions for action descriptions.**

- "Please identify the primary object in the first-person view and describe the main actions involving it."
- "Determine the main object and outline the key actions associated with it."
- "First, identify the primary object, then summarize the main actions performed with it."
- "Locate the primary object and describe the key actions taken."
- "Focus on the interaction between objects and the human."
- "Follow these two guidelines: (1) identify the main objects, and (2) describe the key steps."
- "Spot the central item and describe the key steps performed."
- "Point out the central object and explain the key steps involved."
- "Focus on the primary item and narrate the sequence of actions associated with it."
- "Do not confuse the objects or hallucinate about them."
- "Answer based on what you observe, without over-interpreting, distorting facts, or fabricating information."
- "Think like a human: first identify the interacting objects, then infer the actions being performed."
- "First, infer the overall action, then identify the category of objects, and finally, use the object category to determine if the action is correct. Please output the final description of the step."

Table 12: **The list of instruction guidelines for action description.**

- "What is the way to [action]?"
- "Can you show me how to [action]?"
- "How can I do [action]?"
- "What is the step to [action]?"
- "Could you explain how to [action]?"
- "What method should I use to [action]?"
- "How should I perform [action]?"
- "What is the best way to [action]?"
- "How would you do [action]?"
- "How to [action]?"

Table 13: **List of instructions for How-to meta-task.**

- "The video is describing the step to [action], is this step completed?"
- "Based on the given video, has the task to [action] completed?"
- "Has the action to [action] completed?"
- "Does the video confirm the completion of the step to [action]?"
- "Has the process of [action] been completed based on the given video?"
- "Has the video shown the completion of [action]?"
- "Based on the given video, has the action to [action] completed?"
- "Has the activity of [action] been successfully completed according to the video?"

Table 14: **List of instructions for Finish-Think meta-task.**

- "This video depicts how to [action]. The current goal is [goal]. What is likely to happen next?"
- "This video depicts how to [action]. What is likely to happen next?"
- "What is likely to happen next?"

Table 15: **List of instructions for Next-Step meta-task.**

| Meta-Task | General Prompts |
|---|---|
| Action-Description | Question: {question}. Please provide a brief description in one sentence. The response should be clear and to the point, containing key action words such as objects, verbs, objects, location, and destination. Avoid unnecessary details or explanations. Please briefly describe the key action in a few words. |
| Finish-Think | This is a " Finish-Think" task where you need to predict if a step is completed or not. Question: {question}. Please answer with either " yes" or " no" . |
| How-To | This is a " How-to" task where you need to explain how to accomplish a specific task. Question: {question}. The response should be clear and to the point, containing key action words such as objects, verbs, objects, location, and destination. Avoid unnecessary details or explanations. Please briefly provide a simple step description in a few words. |
| Next-Step | This is a " Next-Step" task where you need to predict the next step in a sequence of steps. Question: {question}. The response should be clear and to the point, containing key action words such as objects, verbs, objects, location, and destination. Avoid unnecessary details or explanations. Please briefly describe the next step in a few words. |

Table 16: **Model inference prompts used for four meta-tasks.**

| Model | Prompts for Evaluation |
|---|---|
| GPT-4O | [Instruction] You are tasked with evaluating the quality of the response provided by an AI assistant. The evaluation should focus on **correctness**, **helpfulness**, and **relevance**. Depending on the task type, you will evaluate specific attributes of a step-level generation tasks or score a simple yes/no question.

**1. For step-level generation tasks**, evaluate the assistant's response based on the following attributes:

**Object**: Does the assistant mention the same or a closely aligned object as the reference? Minor but relevant differences (e.g., an additional unnecessary object) can receive partial credit, but introducing unrelated or missing key objects should lower the score.

**Action Type**: Is the action in the assistant's answer precise and in line with the reference? If the intent of the action is similar but less precise, give partial credit. However, if the action significantly changes the task's context or result, it should be more strictly penalized.

**Location**: Does the response correctly identify the location or context of the action? If the action involves movement (e.g., moving an object from one place to another), evaluate if the destination, starting point, or path are accurately described. Minor location discrepancies can receive partial credit, but if the location changes the context or goal of the action, assign a lower score.

**Attribute**: Are the attributes of the object(s) (such as size, color, state, or condition) correctly described? Missing or incorrect key attributes should lead to a lower score. If attributes are implied but still align with the context, partial credit can be given.

**Scoring**: If the reference answer does not include information for a particular attribute (e.g., object, action type, location, or attribute), do not score that attribute. For each attribute, assign:
- 1 if fully correct,
- 0.5 if somewhat correct or partially aligned,
- 0 if incorrect.

After evaluating each attribute, sum the scores and calculate the overall rating by averaging the individual scores. **Do not round the final result**. The final rating will be a non-rounded average score between 0 and 1.

**2. For yes/no questions**, directly evaluate whether the assistant's response is correct: Assign 1 if the answer is correct, and 0 if it is incorrect.

After providing your analysis, rate the response with the calculated average score, formatted as: **"Rating: [[average_score]]"**. Now proceed with the evaluation based on the provided task:

[Task Type] {task_type},
[Question] {question},
[The Start of Reference Answer] {refanswer},
[The End of Reference Answer],
[The Start of Assistant's Answer] {answer},
[The End of Assistant's Answer]. |

Table 17: **Model inference prompts used for four meta-tasks.**

