# OpenReview forum: "EVA: An Embodied World Model for Future Video Anticipation"
_ICLR.cc/2025/Conference — Submitted to ICLR 2025_

### Official Review · Reviewer_ECZA · 2024-11-01

**Soundness:** 2
**Presentation:** 3
**Contribution:** 3
**Rating:** 5
**Confidence:** 5

**Summary:**

This paper introduces a unified framework for video understanding and generation in embodied scenarios called EVA. The model can take as input an image with an instruction, and autoregressively generate future images until the goal is estimated to be completed. This paper introduces a new benchmark EVA-Bench for evaluation, evaluating video generation quality, the finish-thinking estimation accuracy and action description quality. Language evaluation metrics and image generation evaluation metrics are both used for evaluation.

**Strengths:**

1. The generation performance in both language and video level is significantly better than the compared methods.
2. The introduced EVA evaluates four interesting aspects of captioning, finish-thinking, how-to, and next-step.

**Weaknesses:**

1. The proposed method largely follows Pandora (https://arxiv.org/pdf/2406.09455), the most critical modification is its application on embodied data, and the reform of tasks. From a benchmark side, I believe this is interesting, but from a model side, this work seems incremental.
2. For the results in Table 1, I hope the authors can give more explanations on the comparison between EVA and "ChatUniVi-loRA", since these methods are very similar on the action description task. There is a big performance gap between ChatUniVi and EVA, however there is no explanation on this.
3. From Table 3, it is clear that the language part has a huge influence on the final score. It is understandable that LLaVa OneVision and qwen2-vl are not trained using the same data. But why the EVAS-V score is not largely influenced by EVAS-L score? Does it imply that one of these scores is not consistent with human understanding?
4. As for the EVA metrics, the video generation metrics largely follows the metrics in Vbench. However, I do not believe all these metrics should be used in the embodied scenario. For example, the aesthetic quality metric. For the Goal Completion Estimation metric, I believe this metric can be better designed. Now it is only the CLIP feature similarity. However I think this similarity is too semantic level and coarse-grained in the embodied AI scenario. This results in the GCE is almost similar for all models. I think spatial-awareness is critical in the GCE metric in the embodied AI scenario, and CLIP feature does not consider this. In the same embodiment with similar object presence, the CLIP features tend to be similar. I believe at least DINO features should be used.

**Post discussion comments.**
Many thanks to the authors' responses in the discussion phase. After the discussion, my concerns for this work remain. The authors admit the lacks on the model side but fail to downweight it in the revision. This is not good, especially for a  paper with an emphasis on the model in its title. As for the benchmark, there remain concerns on the evaluation metrics. Using misleading metrics for a new benchmark is not the responsible way to contribute to the research community. Since the contributions in this work is also clear, I hope the authors can carefully re-draft and meticulously re-design the benchmark metrics, to make this work in a mature form to appear in a top-tier conference.

**Questions:**

Please see weaknesses, especially 2,3 and 4.

---

> ### Author Response · Authors · 2024-11-23
> **Q1 Comparison with Pandora**
>
> Dear Reviewer,
>
> Many thanks for your valuable comments! Here, we will give a point-by-point response to your comments. We are also working on supplementing more experiments and updating our manuscripts later according to your advice.
> ## Q1
> First of all, thanks for your appreciation of the task and benchmark. We also noticed that we share a common framework with recent all-in-one frameworks, Next-GPT, miniGemini, etc, especially Pandora. Compared with Pandora, although we both use VLM+Generation, Pandora targets general scenarios and demonstrates instruction for video generation by a generation adapter. Similar to other all-in-one frameworks, they target using large-scale data to align two models.
>
> In embodied generation tasks, such a straightforward alignment method can cause conflicts between different embodied scenes due to the unbalanced scale of different datasets and the fact that embodied cases are not well-studied by pre-trained models. For example, when we follow the setting of Pandora, we find that the robot hand appears in ego-centric scenes( more comparisons can be found in Section D in the anonymous link https://sites.google.com/view/iclr-eva-rebuttal , baseline w/o E-LoRA is similar to Pandora). Therefore, we proposed the E-LoRA and multi-stage alignment methods which are adaptive to robot scenes and also improve the model structure. Moreover, we introduce a QA-based generation, which is different from the instruction-to-video generation other work can do.
>
> In summary, the differences are:
> 1. Different token design for text&video to text&video interleaved generation.
> 2. E-LoRA as a task-specific design under the unbalanced embodied data distribution.

---

> ### Author Response · Authors · 2024-11-23
> **Explaination of EVA and “ChatUniVi-loRA”**
>
> ## Q2
> Many thanks for pointing out the unclear description of this part.
>
> Here, we provide a more comprehensive explanation and will update our manuscript accordingly. The 'ChatUniVi-LoRA' and 'ChatUniVi-Full-Parameter' models are directly fine-tuned on a 50K subset of our EVA instruction tuning dataset using the pre-trained checkpoints provided by ChatUniVi, in order to compare the effects of different training methods.
>
> In contrast, 'EVA' follows a complete multi-stage training pipeline: first aligning the VLM adapter by pretraining the projection layer, then fine-tuning both the projection layer and the full parameters of the LLM using an instruction tuning dataset that combines web-scale data with the embodied dataset. We found that mixing the general VQA and embodied VQA datasets yields better training results.
>
> In summary, compared to the 'ChatUniVi-LoRA' and 'ChatUniVi-Full-Parameter' models, the significant improvement of EVA arises from better alignment and the infusion of extensive general VQA knowledge into the embodied scenes.

---

> ### Author Response · Authors · 2024-11-23
> **More comparision experiment about the  Goal Completion Estimation.**
>
> ## Q4
>
> Again, we would like to show our respect for your valuable suggestions. We were working on updating the benchmark design of EVA, the results are shown in the following table and will be updated in the manuscripts.
>
> We employed the common method of calculating cosine similarity using the last hidden state of DINOv2 to obtain similarity scores. We compared the last frames of generated videos with those of ground truth videos to assess Goal Completion Estimation. Utilizing DINOv2's strong spatial perception for 2D images, we evaluated the embodied videos. Since the datasets for embodied scenes often contain similar scenarios, the observed images are typically quite alike, only with variations mainly in object quantity, appearance, color, and light. Therefore, the DINO score can not separate the difference between each model, as shown in the table, the model with the same generator variance within 1 point.
>
>
> Therefore, we referenced [1] and used DreamSim as a metric, which is tuned to better align with human perception using synthetic image triplets (composed of a reference frame and related frames A and B). DreamSim determines which image, A or B, is most similar to the reference, thereby bringing the distances of similar features closer together. This metric emphasizes foreground objects and semantic content while being sensitive to color and layout. We selected DINOv2_ViTB16 and DINOv2_ViTB14 as the visual backbones, and the results are presented in the following Table. The results aligned with previous experiment results, but had larger variance. In this score, the EVA-Generator achieves the best performance.  In the model that take video as input, EVA and EVA-2Stage also obtain best performance.
>
> | Model                        | Input       | Dreamsim$\downarrow$ | DINO$\uparrow$ | CLIP-I $\uparrow$ | FVD$\downarrow$ |
> |------------------------------|-------------|----------------------|----------------|-------------------|-----------------|
> | Dynamicrafter~\citep{xing2023dynamicrafter} | Image+Text  | 32.36                | 84.67          | 80.96             | 362.56          |
> | Dynamicrafter-Tune           | Image+Text  | 16.32                | 92.35          | 80.72             | 235.52          |
> | EVA-Generator                | Image+Text  | 14.09                | 93.49          | 86.83             | 177.28          |
> | ChatUniVi+Dynamicrafter      | Video       | 35.78                | 82.84          | 80.80             | 314.11          |
> | qwen2-vl-7b+Dynamicrafter    | Video       | 35.65                | 83.18 | 81.96             | 307.33          |
> | ChatUniVi+EVA-Generator      | Video       | 17.21                | 90.95          | 88.48             | 189.61          |
> | qwen2-vl-7b+EVA-Generator    | Video       | 16.48                | 91.58          | 85.87             | 193.89          |
> | LLAVA-OneVision+EVA-Generator| Video       | 18.08                | 91.29          | 88.74             | 192.83          |
> | EVA-2Stage                   | Video       | **15.58**            | 91.93 | **90.19**         | 185.89 |
> | EVA                          | Video       | 16.04    | **92.20**      | 89.09 | **184.81**      |
>
> [1] Fu, Stephanie, et al. Dreamsim: Learning new dimensions of human visual similarity using synthetic data. arXiv preprint arXiv:2306.09344 (2023).

---

> ### Author Response · Authors · 2024-11-23
> **Why EVAS-V score is not largely influenced by EVAS-L score?**
>
> ## Q3
>
> We would like to discuss this point in two aspects, the benchmark score design and the model effect.
>
> From the benchmark design side, we proposed strict rules to evaluate the description ability. Such ability in generation will not reflect as seriously as in language scores. Moreover, the EVA-S considers video generation quality a lot, which would also result in an unbalanced. These quality scores might be important for other post-process (etc. Gesture capture in Sec. C of anonymous linkshttps://sites.google.com/view/iclr-eva-rebuttal/), but not directly reflect the text instruction following ability.
>
> From the VDM model side, the 'EVA-generator' already includes the strong ability of image-to-video animation, even without text instruction, our foundation model can still animate a video (we inject the image condition by contacting it with the text condition). The VDM can give a proper generation result with less strict text instruction, however, to achieve the best performance, structured instruction is still necessary(in Tab.3 NEXT-STEP task, EVA+Generator is about 10+ better than X+Generator).
>
> In summary, we define the embodied video prediction as a multimodal understanding and generation task. Therefore, both scores are important for this task, while their inner connection to them might reflect the model structure, pre-trained data distribution, and alignment quality. Therefore, such an imbalance, on the contrary, aligns with human intuition.

---

> ### Author Response · Authors · 2024-11-25
> **Further discussion to Reviewer**
>
> Dear Reviewer ECZA,
>
> We extend our heartfelt gratitude for your exceptional efforts and insightful comments during the review of our paper. In response to your concern, we discussed them point by point, especially the extra metrics discussion based on your comment. With the discussion phase drawing to a close in the next few days, we eagerly await your feedback and reflections on our responses. We sincerely hope that you will take into account our reply in your evaluation. Should there be any points requiring further clarification or lingering concerns, please be assured of our dedication to promptly address them and furnish you with a thorough response.
>
> Warm regards,
>
> All Anonymous Authors

---

> ### Comment · Reviewer_ECZA · 2024-11-27
>
> Thank you for the response. For the first point, as the authors say in the response, I believe my point still holds. In terms of model design, this work shares a very similar architecture with Pandora.
>
> For the second point, I am not quite following the authors response. Does the response mean that in Table 1, the 'ChatUniVi-LoRA' and 'ChatUniVi-Full-Parameter' models are not fine-tuned using the same data as EVA? If so, it becomes hard to determine whether the data or the training scheme makes the most contribution.
>
> The author's response does not fully convince me of point 3, but I will not consider it as a weakness in my final rating.
>
> For the fourth point, I think the DreamSim may not still be ideal for this scenario, but indeed is much better than a CLIP score. This is because DreamSim is actually robust to transforms such as flipping, However flipping should be considered extremely different in the embodied AI cases. Also, I do not see an updated part related to this point in the revised paper. I would hope more analysis of the different performances regarding the image-based and video-based models.
>
> For the fourth point, I think the DreamSim may not still be ideal for this scenario, but indeed is much better than a CLIP score. This is because DreamSim is actually robust to transforms such as flipping, However flipping should be considered extremely different in the embodied AI cases. Also, I do not see an updated part related to this point in the revised paper. I would hope more analysis of the different performances regarding the image-based and video-based models.

---

> > ### Author Response · Authors · 2024-11-27
> >
> > Reviewer ECZA,
> >
> > Thanks for your feedback.
> >
> > From the model side, We strongly believe that all VLM+VDM papers share a similar structure, If you have time, you are welcome to review more related work, etc survey[1], Video-lavit[2], miniGemini[4], miniGPT5[3], Videodirectorgpt[5] .... in which, Pandora is just one of these good works.
> >
> > From our prospect, we admitted that the model structure is not our main contribution. The contribution of this paper includes the self-ask inference pipeline, task definition, benchmark design, and task-specific design, and we further add extra experiments on simulation robot environments. Moreover, we have shared the failing cases while we training w/o E-LoRA as Pandora on the demo page, if you have time, please check the result.
> >
> > Please do not overlook the key points of this work, and limit the discussion to the difference of model. For example, your advice on the evaluation metric is valuable, and we add the extra comparison of DINO features. Do you have more suggestions or discussions on these parts?
> >
> > [1] He, Yingqing, et al. "LLMs Meet Multimodal Generation and Editing: A Survey." arXiv preprint arXiv:2405.19334 (2024).
> > [2] Jin, Yang, et al. "Video-lavit: Unified video-language pre-training with decoupled visual-motional tokenization." arXiv preprint arXiv:2402.03161 (2024).
> > [3]Zheng, Kaizhi, Xuehai He, and Xin Eric Wang. "Minigpt-5: Interleaved vision-and-language generation via generative vokens." arXiv preprint arXiv:2310.02239 (2023).
> > [4]Li, Yanwei, et al. "Mini-gemini: Mining the potential of multi-modality vision language models." arXiv preprint arXiv:2403.18814 (2024).
> > [5]Lin, Han, et al. "Videodirectorgpt: Consistent multi-scene video generation via llm-guided planning." arXiv preprint arXiv:2309.15091 (2023).

---

> > ### Author Response · Authors · 2024-11-28
> > **Dear Reviewer ECZA**
> >
> > Dear Reviewer ECZA
> >
> > Thanks to your feedback again for your qualitative and valuable advice on the GCE metric. We have updated the experiment and the appendix, where we use the DreamSim for evaluation now. As you mentioned, there is much room for research on the Goal Completion Estimation, from temporal, special, physical, and many other domains. While surveying we noticed that there is still a lack of methods to evaluate a task completion metric from the pixel level, and we believe this might also be an insightful area to discover. Though the discussion period is closing, we will keep testing more image-based and video-based models according to your advice.
> >
> > ## As for the second point
> > We noticed the unclear description and comparison of 'ChatUniVi-LoRA' and 'ChatUniVi-Full-Parameter', and EVA during the rebuttal. As you are right, they are two separate groups, under different training settings. The comparison results only aim to support one point:  Full-Parameter is better than LoRA on our dataset, therefore EVA tunning all parameters. Compared to full-parameter finetunning on the 50K subset, Adding more data works well.
> >
> > ## As for the third point
> > We could discuss it from another angle. In short, Language and Vision scores are positively correlated but aren't strictly linear (like ( y = x )). Instead, it's more nuanced and step-like, reflecting the complexity of how these models interpret and generate content based on varying conditions. Their related curve is closer to a ladder shape (according to different VDM): As the strong image condition injunction, and during VDM training, the language condition is often randomly dropped. Therefore, when the language instructions are not precise enough, the image-to-video model's generation results mainly depend on the image.  The EVA-Generator is a strong image-conditional video generation model. We further tested the EVA-V without language condition and it still reached 46.85 in EVA-S, and the ground-truth prompts lower than 70. Therefore, this distribution is affected by the VDM heavily, which results in limiting the EVA-L's significance in EVA-S.
> >
> > However, when the language instructions are more concise and the verbs are clearer, and the VDM becomes more powerful, the overall prompt capability and the final EVA-S will rise to a higher level.
> > For example, "move forward," "ahead," and "forth" are synonyms that are used by different VLMs. But for VDM, they can only understand the concept of "move forward." Thus, although the language might be very similar, it is difficult for VDM to understand and generate similar videos. This can only be fixed by 1. VDM becomes more general, or VLM learns to use the correct word.
> >
> > Last, we would like to express our respect again to reviewer ECZA. Since video prediction for the world model for Embodied AI is a new task, valuable comments are very helpful to promoting better research.

---

> > > ### Comment · Reviewer_ECZA · 2024-12-03
> > >
> > > Thanks to the authors for the feedback. As I said even in the "weakness" part, I totally acknowledge the contribution of this work. Letting alone the model side, I believe the benchmark provides interesting contributions like new task definition. However, I find the current benchmark is not mature in its form. While the new tasks are introduced, the evaluation of these tasks needs to be more carefully designed. So while the authors convinced me in the response that the excessive emphasis on model design is not necessary, my final rating remains not changed.

---

> > > > ### Author Response · Authors · 2024-12-03
> > > >
> > > > We still thank reviewer ECZA, it was a joyful discussion. Your comments helped us improve the quality of the paper. However, as you mentioned, you acknowledge the contribution of this work, so it might be "marginally above" rather than "below" in the ICLR standard. Many thanks again, even if there is still no room to raise the score.

---

> > > > > ### Comment · Reviewer_ECZA · 2024-12-03
> > > > >
> > > > > I hope the authors do not misunderstand my comments. I can acknowledge both the pros and cons of this work, while I find the paper could be further improved before being published in a renowned venue like ICLR.
> > > > >
> > > > > With the current evaluation metrics (mostly coming from image/video generation works), I believe it is hard to provide a correct estimate of "Finish thinking (goal completion)", while for "how to" and "next step", the evaluation is more reasonable than "goal completion", however, there are still ways that models can hack these metrics.
> > > > >
> > > > > If these points are adequately addressed, I believe this work is very valuable to the community and should be highlighted accordingly. However, I think in its current form, there is the risk that the benchmark provides a vague direction for following future works.

---

> > > > > > ### Author Response · Authors · 2024-12-03
> > > > > >
> > > > > > Just for more discussion, We did show that the current finish thinking is positively correlated with the task completion rate on robot manipulation tasks.
> > > > > >
> > > > > > ### ** Response Tab. Quality comparison on CALVIN **
> > > > > > | model\task | lightbulb | led | rotate | Open drawer | Total |
> > > > > > |------------------------------|---------|---------|---------|----------|----------|
> > > > > > |EVA* (w/o finish thinking)| 12/41 | 43/54 | 98/157|18/41| 171/293(57.24%)|
> > > > > > |EVA* | 35/41| 45/54|  124/157|36/41| 240/293(81.35%) |
> > > > > > |EVA (w/o finish thinking)+ConditonalMLP| 7/41| 13/54|  32/157| 4/41| 44/293(15.01%)|
> > > > > > |EVA+ConditonalMLP | 9/41| 15/54|  45/157| 9/41| 83/293(28.32%)|
> > > > > >
> > > > > > "while for "how to" and "next step", the evaluation is more reasonable than "goal completion", however, there are still ways that models can hack these metrics." this is hard to follow, could you give more explanation or examples?

---

> > > > ### Author Response · Authors · 2024-12-03
> > > >
> > > > Evaluation of the embodied task by pixel is a valuable open question. A recent 1x-world-moidel[1] technical report also highlighted the complexity and released a related challenge of this task. As they mentioned "Evaluation. This is our ultimate goal: can you predict how well a robot will perform before testing it in the real world? "[1].  In our paper, and the rebuttal discussion, we have tried a few possibilities and steps forward to solve this problem, but this does not mean that we can close this task. For the good of the community, we also hope more researchers notice this potential task.
> > > >
> > > > [1] https://www.1x.tech/discover/1x-world-model-sampling-challenge

---

### Official Review · Reviewer_7D8x · 2024-11-04

**Soundness:** 4
**Presentation:** 4
**Contribution:** 3
**Rating:** 6
**Confidence:** 5

**Summary:**

This work proposed a a unified framework called Embodied Video Anticipator (EVA) for both video understanding and generation. A  video generation model was embedded into a VLM model. Comprehensive testing on EVA-Bench demonstrates EVA’s capability to greatly enhance performance within embodied environments, signaling a strong foundation for deploying large-scale pre-trained models in real-world predictive applications.

**Strengths:**

1.  Well motivated: Two main challenges, lack of benchmark and failure in the in embodied scenes , are identified.  As a result, 4 meta tasks are unified into the a single framework to deal with multi-level prediction.

2.  Readability; Well written. Easy to follow. Nice figure.

3.  Originality: A novel world model, the Embodied Video Anticipator (EVA), is designed for video prediction and understanding simultaneously.

4.  New benchmark: Datasets from various domains are aggregated to build a diverse dataset and introduce a multi-stage pretraining approach.

5.  Good performance: EVA outperform all baselines on various tasks and benchmarks.

**Weaknesses:**

1.  Ensemble LoRA: Why is the ensemble LoRA injected into the spatial part? If it’s responsible for the task-specific domain, wouldn’t it make more sense to inject it into the temporal transformer?  Since there is already an image encoder, a more precise explanation of the role of the ensemble LoRA would be helpful.


2.  A further analysis of the output of the gating function would be more helpful to demonstrate the effectiveness of the gating mechanism and to prove efficient adaptation.


3.  Embodied Video Prediction: Embodied Video Prediction involves not only first-person perspective videos but, more importantly, egocentric motion, which this work does not seem to explore. If there are examples related to generated videos or tasks in this area, it would be great to have them updated on the website.

4.  Ablation study:  It would be more helpful to see more ablation studies to verify the effectiveness of each newly added module or method.

**Questions:**

All of my questions are asked in the weakness part

Typo: line 305 Ensamble---Ensemble

Figure 4 Resempler---Resampler?

---

> ### Author Response · Authors · 2024-11-24
> **More explaination of Ensembgle LoRA**
>
> Dear Reviewer,
> To begin with, we first appreciate your response, which would help us improve the quality of this work. We will fix the typo on the manuscript, and answer the weakness point by point. Here we are giving the explanation of Weaknesses 1 and 2, and we were working on giving more experiment results for the ablation study of the added module and gating system and egocentric motion.
>
> ## Q1
>
> Thanks for pointing out the less clear writing part of this module. We inject the LoRA components into all attention blocks in VDM. As we described in Line 303, all temporal and spatial blocks have LoRA. We specifically highlight the spatial part because it includes more detailed condition injections. We will update Figure 4 to have a clear definition.
>
> ## Q2
>
> In the ablation study of the gating system, we compare our model(EVA-Generator) with the model without Ensemble-LoRA that tunes all data together(Dynamicrafter-Tune) in Tab2.  As shown in the table, the EVA-Generator sacrificed DD(14.28 lower), but was much better in all other scores, especially GCE(6.11 better) and FVD(58.24 better). Such a huge performance gap comes from the unbalanced data distribution of different embodied scenes, and the Ensemble-LoRA can efficiently solve this issue by separating different scenes in different LoRA.
> Moreover, quality results also support the point. The ego camera motion, and robot hand, will affect each other and result in worse motion. We give two more failing cases of Dynamicrafter-Tune in Section "D. w/o Ensamble-LoRA" in an anonymous link https://sites.google.com/view/iclr-eva-rebuttal/%E9%A6%96%E9%A1%B5

---

> ### Author Response · Authors · 2024-11-24
> **More experiments and discussion about egocentric motion**
>
> ## Q3
>
> It is a good point! We do not include the special model design for ego-motion. However, we include the video data with keypoint annotation in our training data. We specially extract the intersection part of the keypoint and language annotation Ego-Exo4d in our training data (around 10k data clips). We believe this will be an insightful task for future research.
>
> To further analyze egocentric motion, we propose leveraging the hand modeling framework HaMeR [1] to enhance the portions of our video where hand movements occur. We consider these motions critical for first-person view videos, as they represent a fundamental distinction between egocentric and other types of videos. Unlike other viewpoints, egocentric videos uniquely capture the fine details of hand movements, making them a key focus for analysis.
>
> To achieve this, we utilize HaMeR to reconstruct hand movements in our generated videos, enabling a clearer understanding of egocentric motion. Specifically, we process the video input through a ViT (Vision Transformer) with a transformer head to predict the current camera position and the shape of the hand motion. Using MANO [2], we then generate a mesh representation of the hand movements. Following this reconstruction, we assess whether the hand movements align with the requirements of the How-To task, some results are shown in the Section C Egocentric-Motion in this link https://sites.google.com/view/iclr-eva-rebuttal/%E9%A6%96%E9%A1%B5#h.uyiej5yjt382
>
> Furthermore, for more discussion, there are a few methods that can help, adding key-point tokens, adding a special decoder, combining the motion capture in the VLM, together with motion guidance ControlNet for VDM, etc. However, limited by the data scale, we are still in the process of experimenting with these methods in our future work. If you have more advice, could you describe more about the egocentric motion you mentioned, like some related work?
>
> [1] Pavlakos, G., Shan, D., Radosavovic, I., Kanazawa, A., Fouhey, D., & Malik, J. (2024). Reconstructing hands in 3d with transformers. In Proceedings of the IEEE/CVF Conference on Computer Vision and Pattern Recognition (pp. 9826-9836).
> [2] Romero, J., Tzionas, D., & Black, M. J. (2022). Embodied hands: Modeling and capturing hands and bodies together. arXiv preprint arXiv:2201.02610.

---

> ### Author Response · Authors · 2024-11-24
> **Module Explaination and Ablation Study**
>
> ## Q4
> Thanks for your advice! We were working on it. Exiting experiments already include some quantitative results of the components, we will update the experiment description to improve the understanding ability. We separate the ablation into two parts, VLM understanding and VDM generation:
>
> ### Tab.1 Action-Description results in comparison of VLM
> | Model                        | BLEU1↑  | BLEU2↑  | METEOR↑ | Rouge-L↑ | CIDEr↑ | Spice↑ | CLIP↓ | GPT-4o↑ |
> |------------------------------|---------|---------|---------|----------|--------|--------|-------|---------|
> | ChatUniVi-loRA               | 0.3007  | 0.1855  | 0.1054  | 0.3268   | 0.8245 | 0.2213 | 24.89 | 31.94   |
> | ChatUniVi-Full-Parameter     | 0.4105  | 0.1544  | 0.1809  | 0.4416   | 1.9012 | 0.3414 | 25.36 | 38.46   |
> | EVA                          | 0.5735  | 0.5012  | 0.3095  | 0.5873   | 4.0139 | 0.5506 | 24.98 | 62.63   |
>
>  As shown in Tab.1, we compare the data and training strategy of our methods. With our dataset, we finetuned the pretrained checkpoint as 'ChatUnivi-LoRA' or 'ChatUniVi-Full-Parameter', or in comparison with our method 'EVA'.  'ChatUniVi-loRA' fine-tune a pre-trained checkpoint on 50K EVA dataset subset by LoRA, and 'ChatUniVi-Full-Parameter' stunned full parameter in comparison. Moreover, EVA used a multi-stage training strategy and used a mixture of data, which obtained the best performance in most metrics, especially in GPT-4o score.
> The result shows the multistage finetuning and alignment method significantly improves performance in all metrics.
>
> ### Tab.2 Finish-Thinking Video Generation Quality Comparison
> | Model                          | Input       | SC ↑   | BC ↑   | MS ↑   | DD ↑   | AQ ↑   | GCE ↑  | FVD ↓    |
> |--------------------------------|-------------|--------|--------|--------|--------|--------|--------|-----------|
> | Dynamicrafter~[1]              | Image+Text  | 87.25  | 91.91  | 96.72  | 63.33  | 43.57  | 80.96  | 362.56    |
> | Dynamicrafter-Tune              | Image+Text  | 83.49  | 89.70  | 97.87  | 64.28  | 36.70  | 80.72  | 235.52    |
> | EVA-Generator                  | Image+Text  | 95.74  | 95.11  | 99.09  | 50.00  | 41.07  | 86.83  | 177.28    |
>
> In Tab.2, to showcase the efficiency of E-LoRA, we compare the original Dynamicrafter, finetuning the Dynamicrafter on our dataset as 'Dynamicrafter-Tune', and the 'EVA-Generator' with multiple task-specific LoRA. The result shows that  'Dynamicrafter-Tune' significantly scopes the untrained checkpoint. Compared with 'Dynamicrafter-Tune', 'EVA-Generator' is much better, especially in GCE(6.11 better) and FVD(58.24 better). Moreover, we use the text result of EVA-VLM to generate the video by 'EVA-Generator', named as ‘EVA-2stage’, in comparison with ‘EVA’ which uses a generation adapter. The 'EVA' has a better FVD(1.08) and better generation quality( better SC, BC, MS, DD, and AQ).
>
>
> ### Tab.3 How-To and Next-Step Task-Level Generation Evaluation Results on the EVA-Bench(copy from paper)
>
> | Task       | Model                           | EVAS-L ↑ | EVAS-V ↑ | EVA-Score ↑ |
> |------------|---------------------------------|----------|----------|-------------|
> | HOW-TO     | LLAVA-OneVision+EVA-Generator   | 33.81    | 59.99    | 46.90       |
> | HOW-TO     | qwen2-vl-7b+EVA-Generator       | 41.54    | 60.40    | 50.97       |
> | HOW-TO     | **EVA-2Stage**                  | 85.51    | 63.82    | 74.67       |
> | HOW-TO     | **EVA**                         | 85.51    | 69.93    | **77.72**  |
> | Next-Step  | LLAVA-OneVision+EVA-Generator   | 16.75    | 51.75    | 34.25       |
> | Next-Step  | qwen2-vl-7b+EVA-Generator       | 42.99    | 57.23    | 50.11       |
> | Next-Step  | **EVA-2Stage**                  | 73.02    | 62.10    | 67.56       |
> | Next-Step  | **EVA**                         | 73.02    | 65.34    | **69.18**  |
>
> Tab.3 also supports this experiment's results. We fixed the same generator and compared the effect of different VLM, to demonstrate the importance of the VLM.  We can see that  There is a positive correlation between EVA-Language and EVA-Vision. We also compare the 'EVA-2Stage' and 'EVA'. 'EVA-2Stage' has the same VDM and VLM compared with EVA. However, the  'EVA-2Stage' inference does not use the generation adapter, but directly uses the language output as an instruction to generate the video.
> 'EVA' has a better EVA-Score in both the NEXT-STEP(1.62) and HOW-TO(3.05) tasks. 'EVA-2Stage' outperforms other 2-stage methods(X+EVAGenerator), proving the effectiveness of our VLM in this task, and EVA is better than 'EVA-2Stage', showing the good alignment quality in our method. This part is also the ablation study of generation adapter and VLM.

---

> ### Author Response · Authors · 2024-11-26
> **Dear Reviewer**
>
> Dear Reviewer,
>
> We extend our heartfelt gratitude for your exceptional efforts and insightful comments during the review of our paper. In response to your concern, we discussed them point by point, especially the extra metrics discussion based on your comment. With the discussion phase drawing to a close in the next few days, we eagerly await your feedback and reflections on our responses. We sincerely hope that you will take into account our reply in your evaluation. Should there be any points requiring further clarification or lingering concerns, please be assured of our dedication to promptly address them and furnish you with a thorough response.
>
> Warm regards,
>
> All Anonymous Authors

---

> > ### Comment · Reviewer_7D8x · 2024-11-26
> > **reply**
> >
> > I appreciate the clear responses from the authors. My concerns are addressed in the rebuttal, therefore I will keep the rating and increase the confidence score.

---

> > > ### Author Response · Authors · 2024-11-27
> > > **Feed back to Reviewer 7D8x**
> > >
> > > Dear Reviewer 7D8x,
> > >
> > > Thanks for your positive comments on increasing the score and especially thanks for your advice on our work again.
> > >
> > > Best

---

### Official Review · Reviewer_5ya3 · 2024-11-04

**Soundness:** 3
**Presentation:** 3
**Contribution:** 3
**Rating:** 6
**Confidence:** 4

**Summary:**

This work aims to address the video prediction problem at the task level, i.e., generating a long video until task completion based on current observations and instruction, which is more challenging than traditional frame-level video prediction. The authors reformulate the task in a coarse-to-fine manner, similar to the approach in "Human Rethinking." Specifically, they decompose the generation of long temporal videos into multiple tasks, including Action-Description, How-To, Finish-Thinking, and Next-Step, thereby generating multiple short videos that are ultimately combined into a long video. Subsequently, based on these four meta-tasks, the authors collected corresponding datasets and obtained the EVA model by adopting multi-stage and efficient fine-tuning methods. Experiments across various dimensions validate the effectiveness of the proposed method.

**Strengths:**

**Clear Writing and Presentation**

- The paper is well-written and generally easy to follow. The problem it aims to address is indeed a challenge in the field of video generation and the embodied world model. It provides appropriate intuition and effectively builds up motivation where needed, as illustrated in Figures 1 and 2.

**Interesting Decomposition of the Task**

- The author successfully decomposes the task into more specific and actionable components. By employing a self-ask approach, a multi-task model is serially executed at the inference stage to achieve the final goal, which is a simple yet efficient method, as shown in Figure 3.

**Strong Results in the Designed Benchmark**

- EVA demonstrates strong performance compared to existing baselines in the designed EVA benchmark, which encompasses four meta-tasks with their task-specified metrics.

**Weaknesses:**

**Novelty**

- The model's architecture is very similar to [1][2], as both utilize a Vision-Language Model (VLM) paired with a Video Diffusion Model (VDM). Currently, I have not observed any comparison or discussion between them in the paper (maybe the multi-stage pretraining scheme inspired by human rethinking?). Moreover, multi-stage pretraining and cross-attention alignment have become common training techniques in this field. Furthermore, I am unclear about the differences or advantages of the proposed Ensemble-LoRA compared to other existing LoRA-MOE technologies, like [3]; the authors also lack discussion or comparison with these methods in their experiments.

**Limitations in Evaluation Dimensions**

- Although the authors are tackling a very challenging problem, i.e., task-level video generation over long temporal sequences, all quantitative experiments focus on evaluating the four meta-tasks, without conducting quantitative evaluations on the complete videos. Additionally, since this is an embodied world model, the generated robot manipulation videos can be tested in simulation [4][5] or in real world [5]. For example, RoboDreamer [6], which the authors frequently mention, has performed quantitative evaluations on RLBench. In summary, the experimental results predominantly test short videos and lack comprehensive quantitative evaluations from various perspectives on the overarching problem, i.e., long video generation, they aim to solve.

**Lack of Ablation Studies**

- While the authors have proposed numerous methods (such as decomposing the overall task into four meta-tasks and employing various training strategies for fine-tuning), they have not conducted any ablation studies to verify the necessity of these methods or their specific roles within the framework. For instance, there is no discussion on which dataset/meta-task among the four meta-tasks is more critical for the model's performance, or how the omission of the Action-Description data affects the overall model. Again, if the proposed video generation can be applied to synthesized/real robotic manipulation tasks, it would be easier to verify the performance of this method, as well as how each meta-task helps.

[1] Hu A, et al. Gaia-1: A generative world model for autonomous driving

[2] Xiang, et al. Pandora: Towards General World Model with Natural Language Actions and Video States

[3] Chen Z, et al. Octavius: Mitigating Task Interference in MLLMs via LoRA-MoE

[4] Du Y, et al. Learning universal policies via text-guided video generation

[5] Ko P C, et al. Learning to act from actionless videos through dense correspondences

[6] Zhou S, et al. RoboDreamer: Learning Compositional World Models for Robot Imagination

**Questions:**

- More convincing evaluations and analyses of the proposed embodied world model, examining how the four meta-tasks help the robotic tasks in addition to their benchmarks in EVA. It would be helpful to clarify the contributions of this paper.
- Please also point out the novelty of the proposed embodied video anticipator, in contrast to the recent state-of-the-art methods.

---

> ### Author Response · Authors · 2024-11-23
> **W1 and Q2 Novelty: more discussion and comparison of model**
>
> Dear Reviewer,
>
> First of all, we highly appreciate your insightful comments that will help improve the quality of our work. In response to your detailed comments, especially questions 1 and weaknesses 2 and 3, we will gradually update our experiments during the discussion period (ablation study of meta-tasks, ablation study of components, and qualitative results in simulator robot) and also update them in our manuscripts.
>
> ## W1
> Here, we first provide a detailed explanation of your question 2, or weaknesses 1, regarding the novelty and more comparisons of our model structure. We respect that you pointed out the lack of explanation and comparison of our method. We will summarize and update the comparison later in our manuscripts.
>
> Compared with Pandora, although we both use VLM+Generation, Pandora targets general scenarios and demonstrates instruction for video generation by a generation adapter. In embodied generation tasks, such a straightforward alignment method can cause conflicts between different embodied scenes due to the unbalanced scale of different datasets and the fact that embodied cases are not well studied by pre-trained models. For example, when we trained our Pandora, we found that the robot hand would appear in ego-centric scenes. Therefore, we proposed the E-LoRA and multi-stage alignment method (which we will describe in detail later), which are adaptive to robot scenes and also improve the model structure. Moreover, we introduce a QA-based generation, which is different from instruction-to-video generation.
>
> Comparing this with Gaia, a successful autonomous driving world model, the embodied world model still lacks a clear definition. We noticed that the world model of autonomy is better studied than other embodied scenes. This difference might stem from the complexity of robot scenes (autonomous driving is more successful than other embodied cases for the same reason). In Gaia, multiple tokens are introduced for better controllable generation; however, such control signals still lack a systematic definition in other embodied scenes. Specifically, the actions in EVA are much more varied (grasp, push, lift, etc.), while driving mainly includes turning and accelerating. Thus, a key question for the robot video generation world model before introducing fine-grained control signals is: Is the task completed? Is the task correct? EVA aims to solve a more open-domain problem set by first addressing these fundamental issues. Therefore, we introduce several tasks to drive the embodied world model into the future.
>
> In comparison with LoRA and MoE, they are both well-known methods for model fine-tuning. However, they typically target tuning one model rather than integrating VLM+VDM. Our proposed E-LoRA is a simple but efficient way to end-to-end tune the VLM+VDM together with their connecting adapters without complex settings. Our method especially benefits embodied scenes, which have many scenarios and large variances in data size. Training them together can cause conflicts. In EVA, the generation adapter that connects the VLM and VDM expects large data for alignment, while the VDM needs a LoRA for every different scenario. If trained separately, complex data preprocessing is required, along with heavy work in separate training. So, E-LoRA is specially designed for integrating VLM+VDM.
>
> In extension, we believe MoE will be another highly potential elegant tuning framework for VLM+VDM, though this area is still under exploration by the community.

---

> ### Author Response · Authors · 2024-11-25
> **W2. Limitations in Evaluation Dimensions**
>
> We appreciate your valuable feedback once again.
>
> EVA focuses on embodied video understanding and generation tasks. The primary contributions of [4] and [5] center around proposing a method for the video-to-action process. We believe that their approaches can utilize the information from these videos to drive robots, however, it remains open for exploration and will be a key focus of our future work.
>
> Here we present more evaluation results for RT1 and CALVIN. Quantitative results have been included in the comments, with visually compelling video results available on the webpage https://sites.google.com/view/iclr-eva-rebuttal. Detailed experimental setups will be updated in the Appendix. We especially compared the "EVA (w/o finish thinking)" with EVA, and performed quantitative evaluations to showcase the longer video generation ability.
>
> ## Comparison in RT1
> We compare the success rates of human evaluation tasks in RT1, following the framework of RoboDreamer[6]. Evaluation spans two groups—seen prompts and unseen prompts—and includes a comparison of AVDC, EVA without finish-thinking, and EVA.
>
>
> ### ** Response Tab.1 Seen task **
> | Model\Tasks           | Pick Object |  Move Object Near Object | Place Object Upright | Knock Object Over|  Open/close |  Place Object into Receptacle | Summary|
> |------------------------------|---------|---------|---------|----------|--------|--------|-------|
> |AVDC |11/16 | 9/12  | 1/2  | 4/4  | 2/8 |  2/8  | 29/50(58%)  |
> |EVA (w/o finish thinking)|13/16 |  12/12  | 2/2 |  4/4 |  4/8   | 6/8  | 41/50(82%)  |
> |EVA |13/16 |  12/12  |  2/2  | 4/4  | 4 /8  | 7/8 |  42/50(84%) |
>
> In seen tasks, we randomly selected 50 tasks from the validation set of RT1, including 6 tasks, across multiple scenes(Pick Object, Move Object Near Object, Place Object Upright, Knock Object Over, Open/close, Place Object into Receptacle). The detailed prompt list will be updated in the Appendix later.
> For the seen tasks, AVDC has a 58% success rate, while EVA is 28% higher in total success rate. Moreover, EVA performed better in the Move Object task with a 100% success rate, showing good prompt-following ability. The Open/close task is especially hard, since a few cases like "open right fridge door" include the transparent glass door.
>
>
> ### ** Response Tab.2 Unseen task **
>
> | Model\Tasks           | Pick Object |  Move Object Near Object | Place Object Upright | Knock Object Over|  Open/close |  Place Object into Receptacle | Summary|
> |------------------------------|---------|---------|---------|----------|--------|--------|-------|
> |AVDC |2/4 | 1/4 | 1/1 | 1/2 | 1/1 | 1/3 | 7/15  |
> |EVA (w/o finish thinking)| 2/4 | 1/4 | 0/1 | 2/2 | 0/1 | 0/3 | 5/15  |
> |EVA |3/4 | 4/4 | 1/1 | 2/2 | 1/1 | 1/3 | 12/15  |
>
> For unseen tasks, we start from the existing cases and manually change the subject or object of the prompt. For example, "Place coke can into the bottom drawer" to "Please close bottom drawer". AVDC performance is better than EVA (w/o finish thinking) in Place, knock, and Open/Close tasks, since the trajectory is longer than EVA (w/o finish thinking) can generate at one time. However, EVA could fix this issue and significantly improve the success rate by keeping extending the video.
>
> ## More Comparison on CALVIN
>
> **Set up:** We use the evaluation set separated by RoboFlamingo[7], and we shared our results on 4 tasks in total. On the supplementary demo page, we share the visualization results of the generated video, and use the generation result for planning. In the Video-to-action model, we build a conditional MLP, taking the predicted video and initial status as input, and giving a 16-frame action prediction. The experiment results are shown in Tab.3.
>
> ### ** Response Tab.3 Quality comparison on CALVIN **
> | model\task | lightbulb | led | rotate | Open drawer | Total |
> |------------------------------|---------|---------|---------|----------|----------|
> |EVA* (w/o finish thinking)| 12/41 | 43/54 | 98/157|18/41| 171/293(57.24%)|
> |EVA* | 35/41| 45/54|  124/157|36/41| 240/293(81.35%) |
> |EVA (w/o finish thinking)+ConditonalMLP| 7/41| 13/54|  32/157| 4/41| 44/293(15.01%)|
> |EVA+ConditonalMLP | 9/41| 15/54|  45/157| 9/41| 83/293(28.32%)|
>
> In this Tab 3, the tag* means human evaluation. In human evaluation, the EVA exceeds the EVA(w/o finishing thinking) with a total success rate of 24.11%. And by the ConditionalMLP, EVA could turn its  34.58%(81.35% to 28.32%) successful cases into a simulation robot. For the ablation study, EVA+ConditonalMLP is also better than EVA (w/o finish thinking)+ConditonalMLP in every task.
>
>
> [4] Du Y, et al. Learning universal policies via text-guided video generation
>
> [5] Ko P C, et al. Learning to act from actionless videos through dense correspondences
>
> [6] Zhou S, et al. RoboDreamer: Learning Compositional World Models for Robot Imagination
>
> [7] Li, Xinghang, et al. "Vision-language foundation models as effective robot imitators." arXiv preprint arXiv:2311.01378 (2023).

---

> ### Author Response · Authors · 2024-11-25
> **W3.1 Ablation Study part 1**
>
> We thank you again for your valuable advice on evaluating our generation model on robot tasks. As described and pictured in Figure 3, the self-ask pipeline, is also named finish thinking. As shown in Response Tab 1,2 we compare EVA with EVA (w/o finish thinking), and show the importance of this self-ask pipeline could significantly improve the generation quality of robot generation tasks, and such improvement would also positively relate to the robot planning task in Response Tab 3.
>
> In EVA, our target is clear and simple: A world model that can give future video predictions of some embodied scenes. So we proposed EVA, a video understanding and generation model, that can generate long videos based on its understanding and reasoning.
>
> Therefore, our target is first to evaluate video generation and understanding separately. From the aspect of a video prediction task, these four meta-tasks sequentially exist. For example, without Action-Description, the self-ask pipeline would not work since the VLM can not answer the question "What am I doing, am I done?", or "What happened, what will happen next?".
>
> Then we discuss their effects on the final video generation together, on the prediction task HOW-TO, and a more complex task that needs reasoning ability NEXT-STEP.

---

> ### Author Response · Authors · 2024-11-25
> **W3.2 Ablation Study Part 2**
>
> ## Ablation study for VLM and VDM
>
> First of all, as a video understanding and generation task, we separately did the ablation study.
> ### ** Response Tab.4 Action-Description results in comparison of VLM **
> | Model                        | BLEU1↑  | BLEU2↑  | METEOR↑ | Rouge-L↑ | CIDEr↑ | Spice↑ | CLIP↓ | GPT-4o↑ |
> |------------------------------|---------|---------|---------|----------|--------|--------|-------|---------|
> | ChatUniVi-loRA | 0.3007  | 0.1855  | 0.1054  | 0.3268   | 0.8245 | 0.2213 | 24.89 | 31.94   |
> | ChatUniVi-Full-Parameter | 0.4105  | 0.1544  | 0.1809  | 0.4416   | 1.9012 | 0.3414 | 25.36 | 38.46   |
> | EVA                          | 0.5735  | 0.5012  | 0.3095  | 0.5873   | 4.0139 | 0.5506 | 24.98 | 62.63   |
>
> In Table 4, we meticulously scrutinize the data and training strategies underpinning our methodologies. Leveraging our dataset, we intricately fine-tuned the pre-trained checkpoints, denoted as 'ChatUniVi-LoRA' and 'ChatUniVi-Full-Parameter', alongside our proprietary approach 'EVA'. Notably, 'ChatUniVi-LoRA' fine-tuned a pre-trained checkpoint on a 50K subset of the EVA dataset using LoRA, while 'ChatUniVi-Full-Parameter' encompassed full parameter tuning in comparison. Furthermore, EVA's utilization of a multi-stage training strategy and a diverse data amalgamation yielded the best performance across various metrics, particularly highlighting the remarkable GPT-40 scores.
>
>
> ### ** Response Tab.5 Finish-Thinking Video Generation Quality Comparison **
> | Model | Input       | SC ↑   | BC ↑   | MS ↑   | DD ↑   | AQ ↑   | GCE ↑  | FVD ↓    |
> |----------|-------------|--------|--------|--------|--------|--------|--------|--------|
> | Dynamicrafter| Image+Text  | 87.25  | 91.91  | 96.72  | 63.33  | 43.57  | 80.96  | 362.56    |
> | Dynamicrafter-Tune| Image+Text  | 83.49  | 89.70  | 97.87  | 64.28  | 36.70  | 80.72  | 235.52    |
> | EVA-Generator| Image+Text  | 95.74  | 95.11  | 99.09  | 50.00  | 41.07  | 86.83  | 177.28    |
>
> In Table 5, we present a comparative analysis of E-LoRA. We juxtapose the original Dynamicrafter with the finetuned version on our dataset, labeled as 'Dynamicrafter-Tune', and the 'EVA-Generator' equipped with multiple task-specific LoRA. The findings underscore that 'Dynamicrafter-Tune' refines the untrained checkpoint. In contrast, 'EVA-Generator' outperforms 'Dynamicrafter-Tune' significantly, particularly excelling in GCE (6.11 improvement) and FVD (58.24 improvement). Furthermore, we leverage the text results from EVA-VLM to facilitate video generation through 'EVA-Generator', denoted as ‘EVA-2stage’, in comparison with ‘EVA’, which employs a generation adapter. Notably, 'EVA' demonstrates superior FVD (1.08) and enhanced generation quality across various metrics such as SC, BC, MS, DD, and AQ, showcasing the effectiveness of E-LoRA.
>
> ## Ablation study for task-level generation
>
> Giving QA instructions, the EVA could respond in the final video through the self-ask rethinking process as shown in Figure 3. Therefore, the EVAS-V here represents the final multi-round generation results and gives an evaluation score to the final video.
> ### ** Response Tab.6 How-To and Next-Step Task-Level Generation Evaluation Results on the EVA-Bench(copy from paper)  **
>
> | Task       | Model                           | EVAS-L ↑ | EVAS-V ↑ | EVA-Score ↑ |
> |------------|---------------------------------|----------|----------|-------------|
> | HOW-TO     | LLAVA-OneVision+EVA-Generator   | 33.81    | 59.99    | 46.90       |
> | HOW-TO     | qwen2-vl-7b+EVA-Generator       | 41.54    | 60.40    | 50.97       |
> | HOW-TO     | EVA-2Stage                  | 85.51    | 63.82    | 74.67       |
> | HOW-TO     | EVA                         | 85.51    | 69.93    | 77.72  |
> |------------|---------------------------------|----------|----------|-------------|
> | Next-Step  | LLAVA-OneVision+EVA-Generator   | 16.75    | 51.75    | 34.25       |
> | Next-Step  | qwen2-vl-7b+EVA-Generator       | 42.99    | 57.23    | 50.11       |
> | Next-Step  | EVA-2Stage                  | 73.02    | 62.10    | 67.56       |
> | Next-Step  | EVA                         | 73.02    | 65.34    | 69.18  |
>
> In Table 6, we explored the impact of different Vision-Language Models (VLM) on a fixed generator, emphasizing the pivotal role of VLM. A positive correlation between EVA-Language and EVA-Vision was observed.
>
> Moreover,  'EVA-2Stage' leverages the same VDM and VLM as 'EVA', it directly employs language output for video generation without a generation adapter. In performance, 'EVA' excelled in EVA-Scores for NEXT-STEP (1.62) and HOW-TO (3.05) tasks. Surpassing other 2-stage methods, 'EVA-2Stage' demonstrated the effectiveness of our VLM, with 'EVA' showcasing superior alignment quality. This section also serves as an ablation study of the generation adapter and VLM.

---

> ### Author Response · Authors · 2024-11-26
> **Dear Reviewer 5ya3**
>
> Dear Reviewer 5ya3,
>
> We extend our heartfelt gratitude for your exceptional efforts and insightful comments during the review of our paper. In response to your concern, we discussed them point by point, especially the extra metrics discussion based on your comment. With the discussion phase drawing to a close in the next few days, we eagerly await your feedback and reflections on our responses. We sincerely hope that you will take into account our reply in your evaluation. Should there be any points requiring further clarification or lingering concerns, please be assured of our dedication to promptly address them and furnish you with a thorough response.
>
> Warm regards,
>
> All Anonymous Authors

---

> ### Author Response · Authors · 2024-11-28
> **Dear Reviewer 5ya3**
>
> Dear Reviewer 5ya3
>
> We have been eagerly awaiting your valuable response.
>
>
> Best Regards, All Anonymous Authors

---

> > ### Comment · Reviewer_5ya3 · 2024-12-01
> > **My concerns are well addressed.**
> >
> > I appreciate the comprehensivce responses from the authors, especially about the ablation studies. Even though the model deisgn is somewhat similar to the prior arts (actually a list of methods share a similar structure but with different claimed motivations...), it is okay if the model is applied in a proper and theoretically sound way. My concerns are well addressed in the rebuttal, and I will raise the rating.

---

> ### Author Response · Authors · 2024-12-02
>
> Dear Reviewer 5ya3
>
> Thanks for your valuable comments and your positive feedback! We have further adjusted the video-to-action function to prove the video generation quality could directly improve the planning success rate. In the anonymous page (https://sites.google.com/view/iclr-eva-rebuttal/%E9%A6%96%E9%A1%B5#h.a05r846axunh ), Section F shows the successful planning on CALVIN. With the improved ConditonalActionMLP module, EVA with simple a video-to-action module also obtains satisfying performance. We compare our EVA+ConditonalActionMLP with the Video2Action model[1]'s baselines, and the result shows that using EVA the high-quality generation video as guidance could obtain a satisfying task execution successful rate, and outperformance all baselines in [1]. Therefore, there is a strong reason to believe that EVA with high-quality video anticipation ability could help high-quality planning.
>
> | model\task | lightbulb | led |  slider left | Open drawer |
> |------------------------------|---------|---------|---------|----------|
> BC*[1]| 47.2 |48.8 |67.2 |36.1 |
> GCBC*[1] |1.6 |38.4 |32.0 |22.4|
> DP BC*[1] |70.4 |79.2|68.8 |56.8|
> DP GCBC*[1] |35.2|44.0|40.0| 17.6|
> |EVA+ConditonalActionMLP| 78.0 | 84.5 | 57.3| 60.9|
>
> [1] Luo Y, Du Y. Grounding Video Models to Actions through Goal Conditioned Exploration[J]. arXiv preprint arXiv:2411.07223, 2024.

---

### Official Review · Reviewer_zBHk · 2024-11-04

**Soundness:** 3
**Presentation:** 2
**Contribution:** 3
**Rating:** 6
**Confidence:** 3

**Summary:**

The paper introduces a framework for video generation and video understanding in embodied scenarios through a world model approach. Inspired by the human thinking, the task of video understanding is broken down into four sub tasks - action description, finish thinking, how-to, and next-step along with video generation. This is introduced as a new benchmark, EVA-Bench.

The world model function takes visual observations and a question as input to produce a predicted video and a text response. The action description consists of five key elements (subject, verb, object, location, destination), finish thinking does a frame level prediction to check if the task is completed, how-to task turns the instruction into a visual output, and next-step compares the predicted action video with ground truth video. The metrics of these sub tasks is named as EVA-Score which considers a weighted sum of language metric and visual contentment metric.

The EVA framework is trained as a multi-stage pretraining network with cross-attention alignment and few shot adaptation with LoRA. The method is evaluated the four tasks on multiple metrics and shows an improvement when compared with the baselines. Overall, the contribution of a world model and dividing the video understanding into four subtasks following human thinking is interesting.

**Strengths:**

1. The paper shows an exhaustive evaluation on multiple metrics.
2. The contribution of designing the video understanding as a world model is interesting.

**Weaknesses:**

1. The subtasks of how-to and next-step have a possibility of various scenarios. How-to task turns the instruction into a visual output and there can be multiple outputs to perform the same instruction. Similarly, in the next-step task, there can be multiple actions that are possible given the actions that have happened until now. How does the model take into account this possibility of various scenarios during training and in metric evaluation?

2. The task of finish thinking which decides if the task has finished on the basis of frame-level prediction might not be accurate in various scenarios. For example, in cooking scenarios, the end frame is not unique and can take up many variations of the same recipe. How does the current approach take into account the different variations of the generated frame?

3. Since the tasks are sequential, error compounding can also occur over time.

**Questions:**

Is there any ordering bias in the next-step subtask? Since there are many possibilities for the next step, does the method develop any bias of the dataset when predicting the next step?

---

> ### Author Response · Authors · 2024-11-24
> **What if there are many possibilities for the next step**
>
> Dear Reviewer,
>
> We are thankful for your positive comments about our work, since this is a new area for discussion, your suggestion helping us a lot in improving the quality of our manuscripts. In the following part, we will point by point a response to the weakness and the question you gave in the last.
>
> ## W1 and Question
>
>
> It is an important question and can be answered by our task definition in the Introduction. We would update the manuscript for a more clear writing.
>
> In short, our EVA aims to give a comprehensive prediction for a task. We do not include how many possibilities for each next-step, but aim to ensure that for each next-step, its video prediction is reasonable and proper.
>
> ### More explaination
> Taking a decision tree as an example, each point would result in multiple next-step predictions. We decompose the complex decision-making problem into its most basic meta issue, as solving the prediction of each task leaf node.
>
> We give the randomness of decision from three levels: 1. frame level; 2. task level, and 3. higher level, as we described in the Introduction.
> 1. The frame level prediction aims at giving an extension of the giving frames, which usually will not cause ambiguity.
> 2. Task level, or language instruction level extension, aims to generate the prediction of the given task. This the what our EVA aims to solve.
> 3. Higher level, is where you mentioned: "how-to and next-step subtasks have various possible scenarios".
>
> For level 3, there should be a decomposition of complex tasks, such as long-horizon tasks, separating the task into different possible meta-tasks (level 2). While EVA, aims at turning these level 2 predictions into video predictions (level 1).
>
> However, due to the limitation of the generation model, the definition of leaf points is still unclear(level 2 to level 1 is not stable). We decompose this 2->1 task into four meta tasks and the rethinking, to ensure the process of level 2 -> level 1 is complete.
>
> ### How do we ensure randomness?
> For the frame level randomness:
> During the training of the How-To task, we aim to leverage the generative model's inherent randomness, akin to general diffusion models, to produce diverse videos. This approach enhances the variability of the videos in simulated action implementations, promoting diversity in the generated outputs. For the metric evaluation of vision, we apply relevant metrics to assess both the quality and diversity of the generated videos, ensuring comprehensive evaluation criteria.
> For the task level randomness:
> In the next-step task training, we fine-tune the Next-Step dataset using ChatUniVi, aligning the large language model to the specific objectives of the next-step task. For metric evaluation, we employ GPT-4 to assess whether the results are reasonable. The VLM could be giving multiple possibilities of next prediction, This allows us to evaluate the generated next steps, even if they do not directly match entries in the dataset, as long as the proposed solution enables the completion of the entire task. This flexibility ensures that the model's performance is evaluated holistically, prioritizing task completion over strict dataset matching.
>
> ### Evaluation Experiments
>
> Moreover, we also include the possibility of various possible scenarios: Given the same observation, there may be different HOW-TO. We gave the qualitative results on the demo page (Action following Generation https://sites.google.com/view/iclr25-eva#h.3q1ny1j40i8k), Figure 7 in the appendix(prompt following the ability of the EVA ), and we were also working on giving more fine-grained result on unseen cases and will show in the link ASAP(Section B Unseen Prompt Results https://sites.google.com/view/iclr-eva-rebuttal/). We will also include the task completion result of the seen and unseen cases( various possible scenarios).
>
> ### Bias of the Order
> Since we target solving the NEXT-STEP only, we do not take the bias of the long-horizon task into our issues. In fact, there is the interesting things and reasons why we need a next-step world model, as a foundation for long-tern prediction.  A brain gives a different possibility, the world model will give the related imagination(video prediction) properly. If the task planning is unreasonable, the simulation result would also become unreasonable. Therefore the brain could avoid this wrong decision.
>
> On the other hand, we tried to keep generating the next-step video. It is also the future target of our work. We share one failing case in the Section E(https://sites.google.com/view/iclr-eva-rebuttal/%E9%A6%96%E9%A1%B5#h.5oz4lifcxltr).

---

> ### Author Response · Authors · 2024-11-24
> **Q2 Task Finish Evaluation**
>
> ## Q2
> A valuable discussion! We include the finished discussion from two aspects and evaluate them by the last frame and VLM.
>
> For the last frame evaluation, we focus on the final frame of the generated video as the primary judgment criterion, with the majority target on solving the robot manipulation tasks. Our reasoning is that the final frame encapsulates the culmination of the task's ongoing process. Assuming the frames are continued, the closer the last frame is, the closer the video process is. This evaluation process is designed to handle the inherent randomness of the diffusion model, which generates variations in the video frames.
>
> In the finished task, we utilize the VLM to evaluate whether the generated video successfully completes the intended How-To task. This involves determining whether the task described in the How-To instruction aligns with the generated video and whether the video achieves the task's completion. This can be especially helpful while evaluating egocentric tasks, which usually have a lot of looping movements(washing hands, cutting carrots, etc.). Moreover, in this case, the last frame evaluation works minor since the motion and the frame distance are also small.
>
> In summary, we have two methods, VLM and last-frame, that remain capable of accurately judging task completion regardless of these variations.
>
>
> Furthermore, in future work, we aim to experiment with using the final frame of the generated video in different steps to explore whether this approach improves the accuracy and consistency of task completion assessments.

---

> ### Author Response · Authors · 2024-11-24
> **Q3 Error Accumulation**
>
> Sure. Error always occurs. Although we proposed a rethinking pipeline and a VDM conditional generation finetuning to minimize the error in the level 2 to level 1 generation process, we still face a lot of difficulty with Error Accumulation.
>
> Error compounding can manifest in two ways:  (1) during the finished task(level 2-> level 1, task to video prediction), where the generated video fails to complete the intended task. or (2) during the next-step task(level 3, higher level prediction and planning), where errors accumulate and prevent the completion of a long-range task, or
>
>
> In the first case, the accuracy of the finished task, as detailed in our paper, serves as a key metric. While instances of error compounding do occur, they also highlight the quality of the diffusion model and its impact on evaluation. These cases provide valuable insights into the limitations and strengths of the model, informing future improvements. Our EVA achieves the best task completion as shown in our experiment, and has high-quality generation, which makes it possible for more future research.
>
> In the second case, addressing error compounding may require a multi-step evaluation framework to assess the sequence of steps generated by the VLM for the task. This would involve evaluating whether the overall task planning aligns with the ground truth (GT). Developing such an evaluation framework represents a direction for future work. However, error compounding is less likely to occur in our current setting due to the robust design of our approach. We also experience the long-horizon task generation, which shows the possibility of future research, like long-term memory of video prediction, in the link(Section E. Failing Cases of Long Horizon Task Generation, https://sites.google.com/view/iclr-eva-rebuttal/%E9%A6%96%E9%A1%B5#h.5oz4lifcxltr)

---

> ### Comment · Reviewer_zBHk · 2024-11-25
> **Acknowledgment**
>
> Thank you authors for thoroughly answering my concerns!
>
> Q1. The decision tree example gives good clarity. The authors are encouraged to explore the various possible scenarios in their future work.
>
> Q2. My concern for the various possibilities of the end frame was also addressed. The authors raised a good point that the final frame is the culmination of the task's ongoing process and because the frames have continuity, the distance and motion would be small.
>
> Q3. The authors addressed my concern about the error accumulation and shared a failure case of their method as well.

---

> > ### Author Response · Authors · 2024-11-25
> > **Further discussion to Reviewer zBHk**
> >
> > Dear Reviewer,
> >
> > Thank you for your timely feedback! As the discussion phase is still in progress, we have included additional quantitative comparisons, such as an ablation study, and further task evaluations to supplement our work. Should you have any further inquiries, please do not hesitate to reach out. If your previous concerns have been satisfactorily resolved, we would be immensely grateful if you would contemplate revising your rating.
> >
> > Best Regards,
> > All Anonymous Authors

---

> > > ### Author Response · Authors · 2024-11-28
> > > **Further discussion with Reviewer zBHk**
> > >
> > > Dear Reviewer zBHk,
> > >
> > > We have recently updated our PDF. If your previous concerns have been satisfactorily addressed, we kindly ask you to consider raising your rating, confidence, soundness, or Contribution.
> > > Thank you for your time and feedback again!
> > >
> > > Best Regards, All Anonymous Authors

---

### Author Response · Authors · 2024-12-02
**Dear all reviewers of EVA**

Dear all reviewers.

We appreciate your valuable comments again! As the discussion period of ICLR2025 only remains less than 24 hours, we here give a quick overview of all our rebuttals. We add a rebuttal demo page(https://sites.google.com/view/iclr-eva-rebuttal) and introduce 5 groups of visualization results, including longer video generation (A), unseen prompt video generation(B)， EVA extension on high-quality Ego Motion(C), ablation of E-LoRA(D), shared failing cases(E), and EVA on robot-simulator(F). We especially thank reviewer zBHk for an insightful discussion on error accumulation, reviewer 5ya3's advice on the robot simulation experiment, reviewer 7D8x's idea on ego motion, and reviewer ECZA's comment about the GCE metrics. This valuable advice has immensely helped improve the quality of manuscripts.


We kindly ask you to have an overall consideration again, if our response has addressed your previous concerns, we would be immensely grateful if you raised your rating, confidence, soundness, or Contribution. If you feel some points are still unclear, please feel free to discuss more in the remaining time.  We believe video generation could become one of the potential approaches to the future embodied intelligence, and all your comments are valuable for approaching this future.

Best

All anonymous authors of EVA

---

### Meta-Review · Area_Chair_iqSz · 2024-12-21

**Metareview:**

This paper proposes a world-model based approach for video-understanding and -generation in embodied scenarios. To this end, the authors introduce both a new benchmark and a model to address this task. Reviewers largely appreciated the overall direction and motivation for this paper.

However, after a lengthy discussion with Reviewer ECZA, some key points remain unaddressed: On the model side, the approach is incremental, and instead of emphasising the model in the paper, it may be more appropriate to highlight the benchmark instead. However, the paper was not revised to reflect this. Moreover, although the benchmark is promising, some concerns remain over its evaluation metrics. It is important for the reserach community to iterate on well-designed benchmarks where evaluation metrics correlate well with actual capabilities of models that users care about. Therefore, the AC agrees with Reviewer ECZA that the paper could do with another round of iteration and submission.

**Additional Comments On Reviewer Discussion:**

Refer to above. The rebuttal featured a lengthy discussion with Reviewer ECZA, and some key points remain unaddressed: On the model side, the approach is incremental, and instead of emphasising the model in the paper, it may be more appropriate to highlight the benchmark instead. However, the paper was not revised to reflect this. Moreover, although the benchmark is promising, some concerns remain over its evaluation metrics

---

### Decision · Program_Chairs · 2025-01-22

Reject